# Developmental YAPdeltaC determines adult pathology in a model of spinocerebellar ataxia type 1

Kyota Fujita[1], Ying Mao[1], Shigenori Uchida[1], Xigui Chen[1], Hiroki Shiwaku[1], Takuya Tamura[1], Hikaru Ito[1], Kei Watase[2], Hidenori Homma[1], Kazuhiko Tagawa[1], Marius Sudol[3,4,5] & Hitoshi Okazawa[1,2]

YAP and its neuronal isoform YAPdeltaC are implicated in various cellular functions. We found that expression of YAPdeltaC during development, but not adulthood, rescued neurodegeneration phenotypes of mutant ataxin-1 knock-in (Atxn1-KI) mice. YAP/YAPdeltaC interacted with RORα via the second WW domain and served as co-activators of its transcriptional activity. YAP/YAPdeltaC formed a transcriptional complex with RORα on *cis*-elements of target genes and regulated their expression. Both normal and mutant Atxn1 interacted with YAP/YAPdeltaC, but only mutant Atxn1 depleted YAP/YAPdeltaC from the RORα complex to suppress transcription on short timescales. Over longer periods, mutant Atxn1 also decreased RORα in vivo. Genetic supplementation of YAPdeltaC restored the RORα and YAP/YAPdeltaC levels, recovered YAP/YAPdeltaC in the RORα complex and normalized target gene transcription in *Atxn1*-KI mice in vivo. Collectively, our data suggest that functional impairment of YAP/YAPdeltaC by mutant Atxn1 during development determines the adult pathology of SCA1 by suppressing RORα-mediated transcription.

[1] Department of Neuropathology, Medical Research Institute, Tokyo Medical and Dental University, 1-5-45 Yushima, Bunkyo-ku, Tokyo 113-8510, Japan. [2] Center for Brain Integration Research, Tokyo Medical and Dental University, 1-5-45 Yushima, Bunkyo-ku, Tokyo 113-8510, Japan. [3] Mechanobiology Institute, National University of Singapore, 5A Engineering Drive 1, Singapore 117411, Singapore. [4] Department of Physiology, National University of Singapore, Yong Loo Li School of Medicine, 2 Medical Drive, Singapore 117597, Singapore. [5] Institute of Molecular and Cell Biology (IMCB) A*STAR, Biopolis, Singapore 138673, Singapore. Kyota Fujita, Ying Mao, and Shigenori Uchida contributed equally to this work. Correspondence and requests for materials should be addressed to H.O. (email: okazawa-tky@umin.ac.jp)

Spinocerebellar ataxia type 1 (SCA1) is a neurological disease that mainly affects Purkinje cells in the cerebellum and motoneurons in the spinal cord[1–3]. It has been more than 20 years since the discovery of the causative gene, Ataxin-1 (*ATXN1*)[4]. Since then, a great deal of knowledge has accumulated regarding the mechanisms of SCA1, including the identities of the Atxn1-interacting factors involved in transcription and splicing[5–9]. However, SCA1 remains intractable, and no disease-modifying therapy has reached the clinical bedside.

Meanwhile, as with other neurodegenerative diseases, our conception of spinocerebellar ataxias (including SCA1) has changed substantially. First, it has become clear that the types of affected neurons are not as specific as originally believed. For example, we now know that SCAs affect not only cerebellar neurons but also cerebral neurons, sometimes leading to cognitive impairment in patients with SCA1[10]. Second, the effects of these disorders are not limited to single organs: polyQ diseases like Huntington's disease affect not only the brain but also muscle, adipose, pancreas, and heart tissues[11], and this may also be the case in SCAs. Third, SCA might be a developmental disorder that affects cerebellar neurons during embryogenesis or in early childhood[12].

In regard to the third point, a study by the Orr group revealed the developmental molecular pathology of SCA1 by showing that Atxn1 indirectly interacts with RORα, an orphan nuclear receptor with similarities to retinoic acid receptors[12]. Expression of mutant Atxn1 protein, encoded by a gene containing a CAG expansion, decreases RORα levels and thus broadly affects expression of target genes necessary for cerebellar development[12].

We previously showed that the transcriptional co-factor YAP is involved in an atypical form of necrosis induced by alpha-amanitin, transcriptional repression-induced atypical cell death (TRIAD) in which C-terminal truncated isoforms of YAP (ins13, ins25, and ins61 possessing additional mini-exon sequences between exon 5 and exon 6) play critical roles[13]. In subsequent experiments, we expressed these isoforms in developing *Drosophila* under the control of the tubulin Gal4 driver, fed the larvae alpha-amanitin, and monitored the survival ratio from larva to pupa. The results revealed that YAPdeltaC-ins61 had the strongest anti-TRIAD activity (our unpublished data). Therefore, we took YAPdeltaC-ins61 (hereafter, "YAPdeltaC") as the representative of three isoforms. Biochemical and morphological data support the idea that TRIAD actually occurs in human Huntington's disease brains[14], and overexpression of YAPdeltaC prevents TRIAD in Huntington's disease in vitro models[15].

In this study, we tested the therapeutic effect of YAPdeltaC on SCA1 pathology using a newly developed Tet-ON YAPdeltaC system in *Atxn1*-KI (Sca1$^{154Q/2Q}$) mice. Unexpectedly, adulthood expression of YAPdeltaC did not ameliorate the pathology and symptoms of *Atxn1*-KI mice. By contrast, YAPdeltaC expression

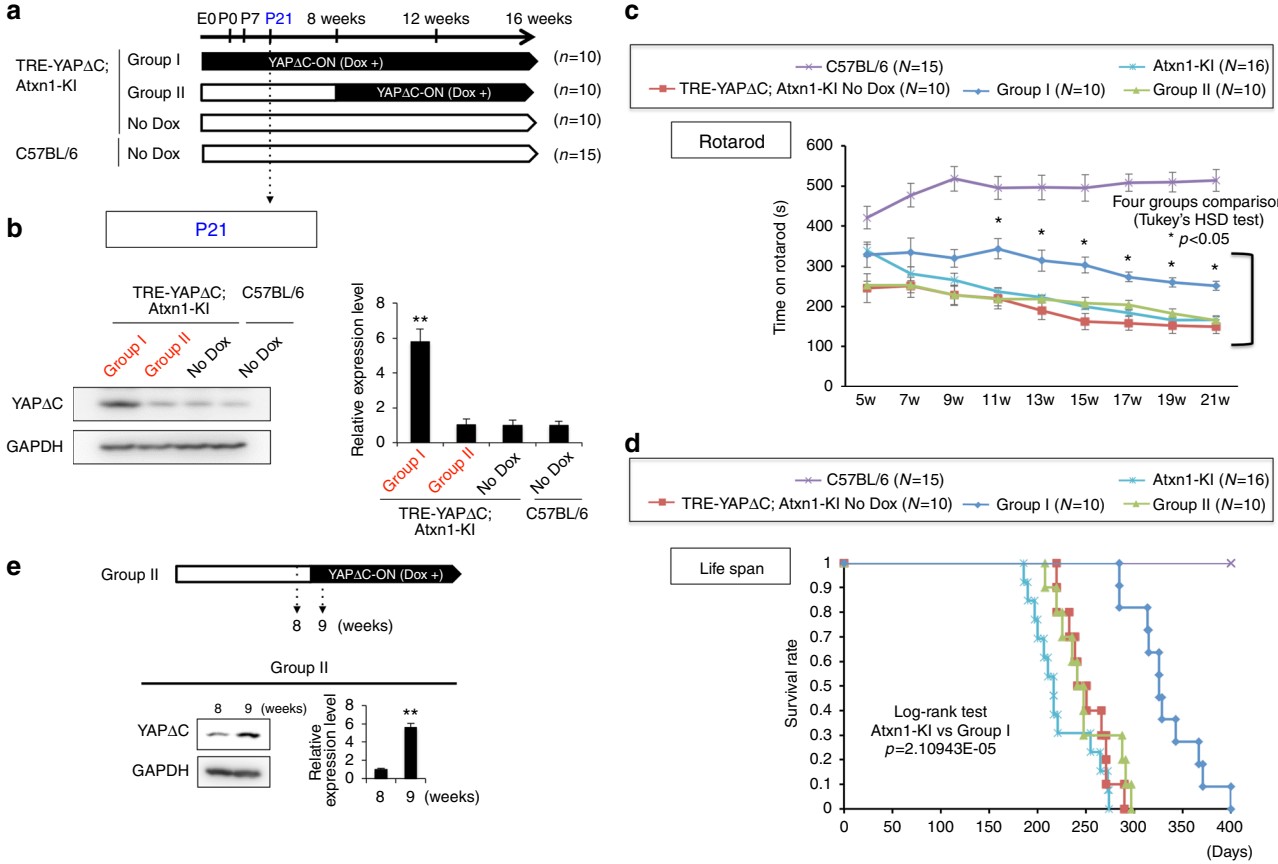

**Fig. 1** Experimental design of YAPdeltaC Dox-ON in *Atxn1*-KI mice. **a** Protocol of Dox feeding of the YAPdeltaC Dox-ON mice. The double-transgenic mice (Tet-ON YAPdeltaC; *Atxn1*-KI) were divided into three groups (Group I, Group II, No Dox) following the Dox administration/feeding protocol as indicated. **b** Protein levels of YAPdeltaC in the cerebellum of Group I mice at P21 were examined by western blot. The blot was re-probed with GAPDH antibodies. Double asterisks indicate statistical significance ($p < 0.01$, $N = 6$) in one-way ANOVA with post hoc Tukey's HSD test. **c** Temporal changes in motor function in five groups of mice, as determined by the Rotarod test. Asterisks indicate significant differences vs. *Atxn1*-KI mice in the multiple group comparison (one-way ANOVA with post hoc Tukey's HSD test, $p < 0.05$). **d** Survival ratio of five groups. Log-rank test confirmed the significance of the lifespan elongation in Group I. **e** Effect of Dox-ON at 8 weeks of age on YAPdeltaC protein expression at 9 weeks of age, evaluated by western blot with YAPdeltaC antibody. Double asterisks indicate statistical significance ($p < 0.01$, $N = 6$) in Student's *t*-test

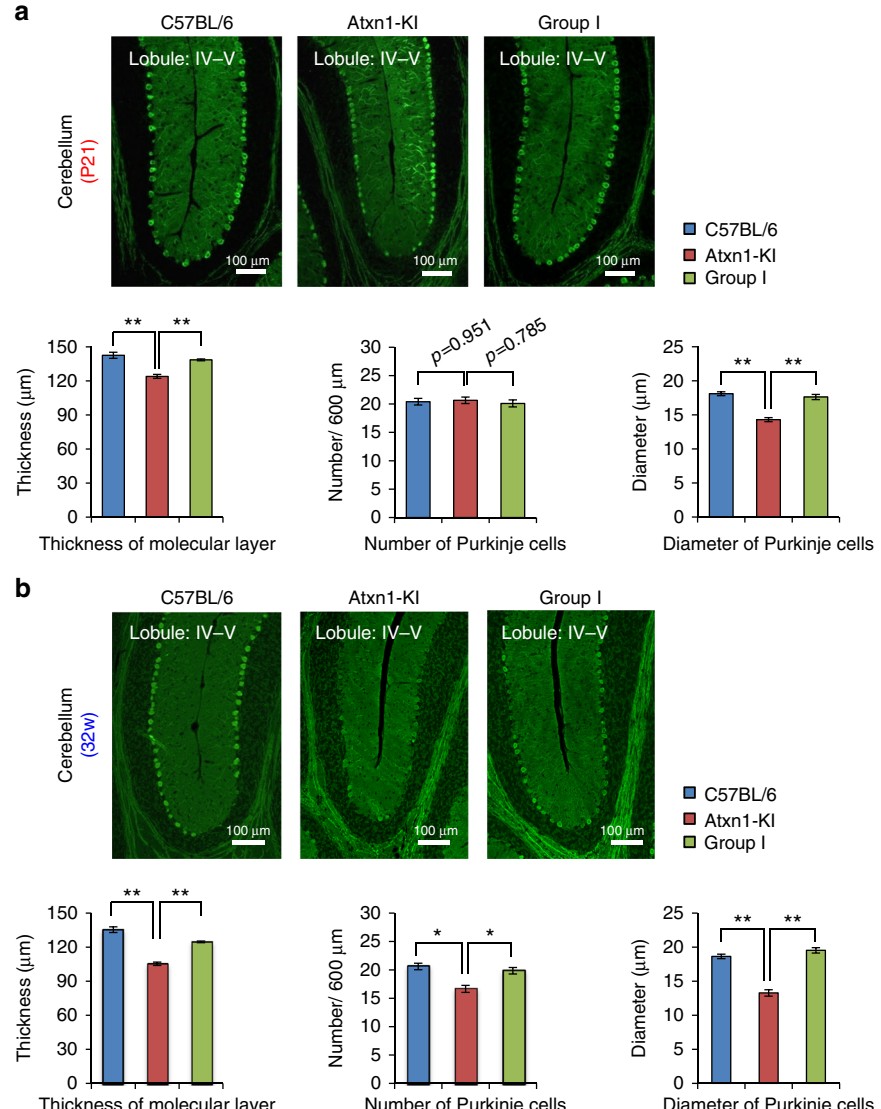

**Fig. 2** Morphological changes precede cell death in Purkinje cells at P21. **a** Upper panels are low-magnification images of the cerebellar cortex in three mouse groups at P21. Sections were immunostained with anti-calbindin antibody. The number of Purkinje cells in 600 μm of the primary fissure side of lobule IV–V was counted in one slide, and the values from 20 slides from four mice (five per mouse) were used for statistical analysis. To determine the thickness of the molecular layer, the thickness was measured at six regions on the primary fissure side of lobule IV–V per slide, and the values from 120 regions from 20 slides from four mice were used for statistical analyses. The diameter of Purkinje cells was measured for all Purkinje cells on the primary fissure side of lobule IV–V, and the values from 20 slides from four mice were directly used for analysis. Lower graphs show quantitative analyses of the thickness of the molecular layer, the number of Purkinje cells, and the diameter of Purkinje cells. Double and single asterisks indicate statistical significance ($p < 0.01$ and $p < 0.05$, respectively) in one-way ANOVA with post hoc Tukey's HSD test. Four mice were used for analyses; 20 slides were made from each mouse; and 50 Purkinje cells were measured in total. **b** Analysis as described for **a**, at 32 weeks of age

during development markedly rescued the pathology and symptoms in adulthood. We found that YAP/YAPdeltaC functioned as a transcriptional co-activator of RORα. Normal Atxn1 collaborated with YAP/YAPdeltaC to activate RORα, whereas mutant Atxn1 inhibited incorporation of YAP/YAPdeltaC into the RORα transcription complex. Supplementation of YAP/YAPdeltaC overcame the toxic effect of mutant Atxn1 and restored the transcriptional activity of RORα. These results elucidate the molecular function of YAPdeltaC in the developmentally determined pathology of SCA1.

## Results

**Developmental YAPdeltaC rescues adult phenotype of KI mice.** We generated transgenic mice expressing YAPdeltaC using the advanced Tet-ON system (see Methods). When fed doxycycline

(Dox), these transgenic mice express YAPdeltaC under the control of the neuron-specific enolase (NSE) promoter. To investigate the temporal specificity of the potential therapeutic effect of YAPdeltaC, the Tet-ON mice were crossed with mutant heterozygous *Atxn1*-KI (Sca1[154Q/2Q]) mice[16]. The double-transgenic mice (Tet-ON YAPdeltaC; *Atxn1*-KI) expressed mutant Atxn1 (Atxn1-154Q) from E0 to death, but expressed YAPdeltaC specifically in neurons only when fed Dox. All mice were of the C57BL/6 genetic background.

In Group I, mother mice were fed Dox during pregnancy and breast-feeding (from E0 to 3 weeks of age); after 3 weeks, the double-transgenic progeny received Dox in drinking water (Fig. 1a). In Group II, the double-transgenic mice received Dox in drinking water from 8 weeks of age until death (Fig. 1a). No Dox group of the double-transgenic mice received no Dox during

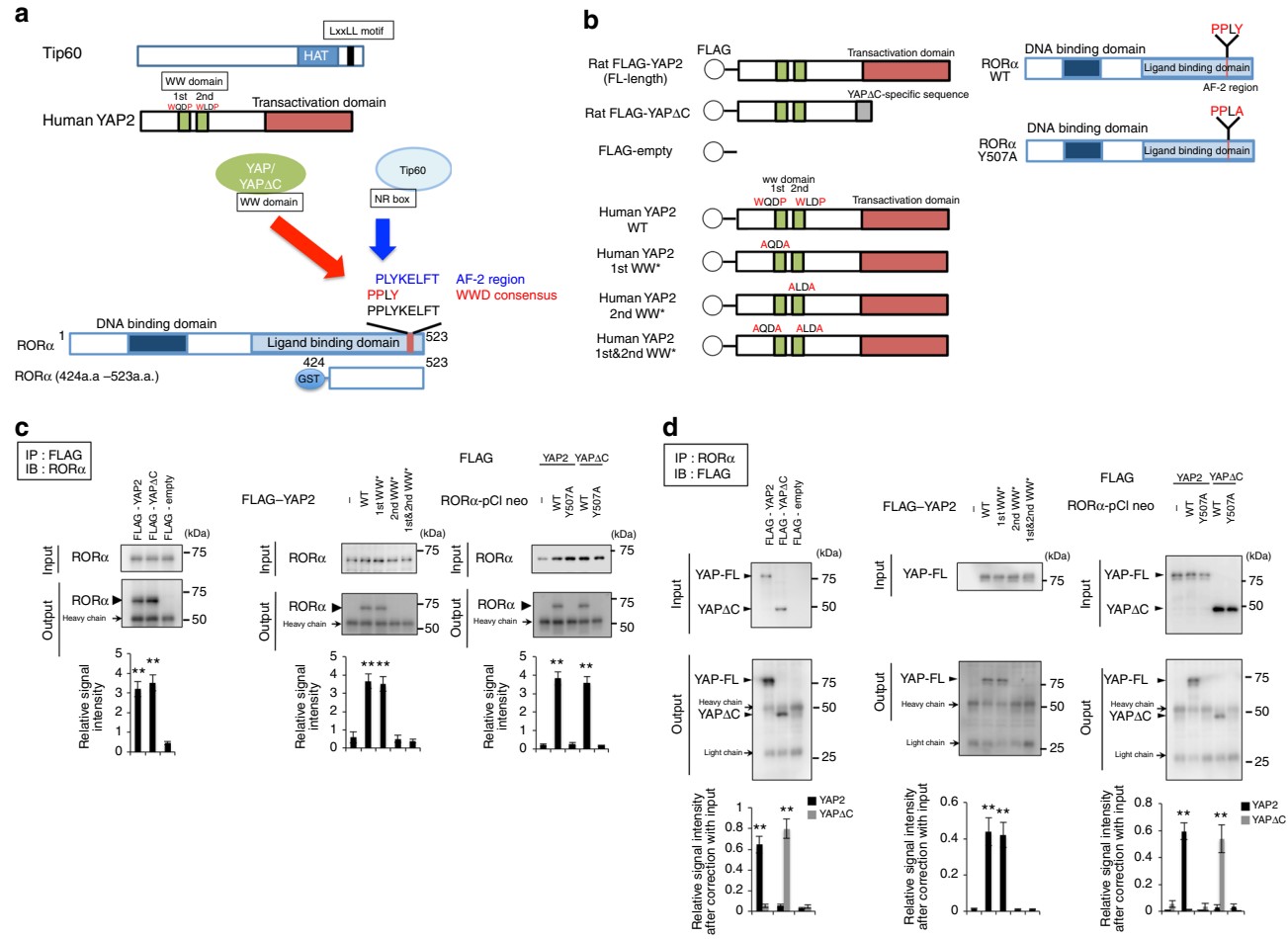

**Fig. 3** Structures of RORα, YAP, and Tip60 and their possible relationship. **a** Scheme showing the interaction among YAP, Tip60, and RORα. GST-RORα (a. a. 424–523), used for the pull-down assay in Fig. 5d, is also shown. **b** Mutants generated for interaction assays. **c, d** In left panels, immunoprecipitation analyses with primary cerebellar neurons transiently expressing FLAG-YAP2 (full-length YAP) or FLAG-YAPdeltaC show the interaction of endogenous RORα with YAP or YAPdeltaC. In middle panels, YAP2 mutants of the first and/or second WW domain were also examined for interaction with endogenous RORα. In right panels, wild-type RORα and mutant RORα at the PPLY motif were tested for interaction with FLAG-YAP2/YAPdeltaC. The lower graphs show quantitative analyses of the relative intensities of output bands, corrected based on the corresponding input band ($n = 3$). Double asterisks: $p < 0.01$ in one-way ANOVA with post hoc Tukey's HSD test

embryogenesis and after birth (Fig. 1a). The induction of YAPdeltaC by Dox in Group I mice was confirmed by analysis of cerebellar tissues from Group I, Group II, No-Dox, and C57BL/6 mice (without Dox feeding) at P21 (3 weeks of age) (Fig. 1b).

The lifespans of the four groups were investigated, and their motor functions were evaluated using the Rotarod test. In *Atxn1*-KI mice, the onset of motor dysfunction was quite early, and Rotarod test scores declined at 7 weeks. In Group II and No Dox, the decline of motor function was similar to that in *Atxn1*-KI mice (Fig. 1). By contrast, the scores remained high in Group I, and the improvement was statistically significant from 11 to 21 weeks of age (Fig. 1c).

Developmental expression, but not adulthood expression, of YAPdeltaC was also effective in lengthening lifespan (Fig. 1d). In the background control group (C57BL/6), mean lifespan was longer than 400 days. By contrast, in *Atxn1*-KI mice, mean and maximum lifespans were 224.1 and 274 days, respectively; in Group I (Tet-ON-YAPdeltaC;Atxn1-KI, Dox+ from E0 to death), 318.3 and 400 day; and in Group II (Tet-ON-YAPdeltaC;Atxn1-KI, Dox+ from 8 weeks to death), 238.4 and 297 days; in No Dox (Tet-ON-YAPdeltaC;Atxn1-KI, Dox−), 238.3 and 290 days.

These differences in lifespan confirmed that the critical period for YAPdeltaC expression is the developmental stage (E0–8 weeks). We confirmed that YAPdeltaC protein was rapidly induced within 1 week after initiation of Dox-feeding at 8 weeks of age (Fig. 1e).

**Developmental YAPdeltaC rescues adult pathology of KI mice.** In *Atxn1*-KI mice at P21, the number of Purkinje cells was not affected (Fig. 2a), but the thickness of the molecular layer was reduced (Fig. 2a). In addition, the diameters of Purkinje cell bodies were significantly reduced in these animals (Fig. 2a). These data obtained from the primary fissure side area of lobule IV–V, which corresponds to the upper vermis severely affected in human SCA1 patients, suggested that differentiation and maturation of Purkinje cells were impaired[12], but cell death did not occur by P21. By 32 weeks, cell number was also decreased (Fig. 2b), indicating that cell death occurred between P21 and 32 weeks.

In Group I mice at P21, YAPdeltaC restored Purkinje cell body diameter and molecular layer thickness (Fig. 2a) and also prevented the decrease in Purkinje cell number at 32 weeks

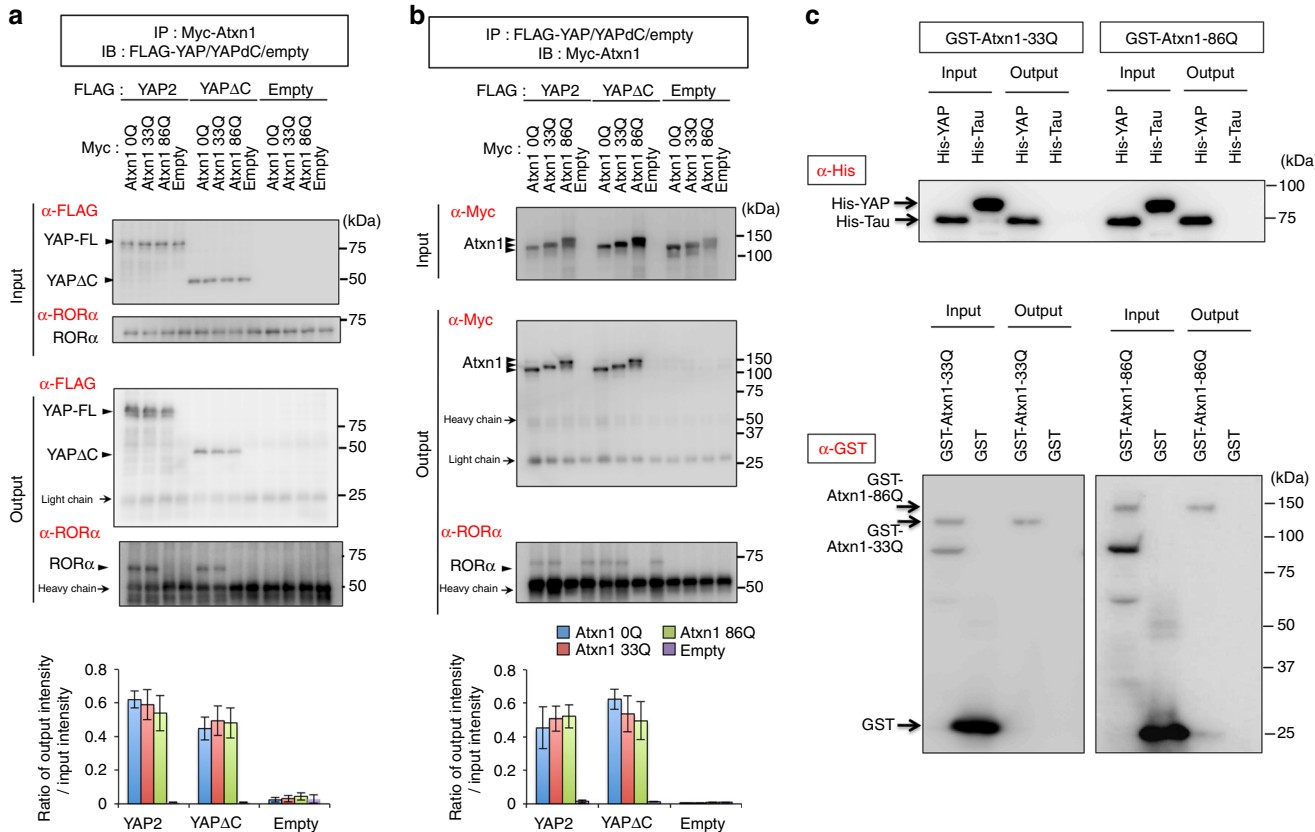

**Fig. 4** Formation of the protein complex containing RORα, YAP, and Atxn1. **a, b** Immunoprecipitation analyses were performed with transient transfection of FLAG-YAP2/YAPdeltaC and Myc-Atxn1-0Q/33Q/86Q in primary cerebellar neurons at 2 days after transfection. The lower graphs show quantitative analyses of the relative intensities of output band, corrected based on the corresponding input band ($n = 3$). Double asterisks: $p < 0.01$ in one-way ANOVA with post hoc Tukey's HSD test. **c** Pull-down assay to test for a direct interaction between YAP and Atxn1-33/86Q. GST-Atxn1 and His-YAP were incubated and pulled down with glutathione–Sepharose (upper panel) or His-resin (lower panels). Pulled-down and interacting proteins were detected with anti-His or anti-GST antibody

(Fig. 2b), in consistence with the late-onset effect of RORα-mediated transcription reported previously[12], and suggested functional interaction between YAPdeltaC and RORα.

**YAP/YAPdeltaC interact with RORα.** Hence, we tested physical interaction between YAP/YAPdeltaC and RORα. We scanned the amino-acid sequence of RORα and found a PPLY sequence that matches the PPxY consensus for interaction with YAP WW domain (Fig. 3a)[17–19]. This PPLY motif also overlapped with a consensus sequence (PLYKELFT) for interaction with Tip60 NR box (Fig. 3a). To determine which YAP WW domain is essential for interaction with RORα, we generated YAP expression vectors mutated at the first and/or second WW domain (Fig. 3b). To investigate YAP binding to the PPLY consensus, we also constructed a RORα mutant in which PPLY was replaced to PPLA (Fig. 3b).

FLAG-tagged full-length YAP or YAPdeltaC was transiently expressed along with wild-type RORα in primary cerebellar neurons and immunoprecipitated by anti-FLAG and anti-RORα antibodies at 2 days after transfection. RORα reciprocally co-precipitated with both full-length YAP and YAPdeltaC (Fig. 3c, d left panels), and deletion of the C-terminal domain of YAP in YAPdeltaC did not markedly affect the interaction (Fig. 3c, d, left panels). FLAG-empty, used as a negative control, did not co-precipitate RORα (Fig. 3c, d, left panels). The interaction was retained when the first WW domain was mutated, but was completely lost in a YAP mutant lacking the second WW domain, as well as in the double mutant lacking both WW domains

(Fig. 3c, d, middle panels). RORα harboring a mutation in the binding motif (Y507A) was not co-precipitated with full-length YAP or YAPdeltaC (Fig. 3c, d, right panels). These results indicated that the second WW domain of YAP/YAPdeltaC and the PPLY motif of RORα are essential for the interaction.

The similar experiments were further performed with COS-7 cells transiently overexpressing RORα at 2 days after transfection, in which endogenous expression of RORα is very low (Supplementary Fig. 1A, B). The results were exactly similar with those from primary cerebellar neurons and supported that the second WW domain of YAP/YAPdeltaC and the PPLY motif of RORα are essential for the interaction.

**Atxn1 forms a complex with YAP/YAPdeltaC and RORα.** Next, we tested the difference between normal and mutant Atxn1 (Fig. 4a, b). From primary cerebellar neurons expressing Myc-Atxn1 and FLAG-YAP/YAPdeltaC at 2 days after transfection, both normal and mutant Atxn1 proteins were co-precipitated with YAP/YAPdeltaC (Fig. 4a, b). Reciprocal co-precipitation of YAP/YAPdeltaC and Atxn1 did not significantly differ as a function of polyQ length (Fig. 4a, b, lower graphs).

In pull-down assays using His-tagged YAP and GST-tagged normal or mutant Atxn1 (Fig. 4c), His-YAP but not His-tagged tau as a negative control was pulled down by glutathione–Sepharose with both GST-Atxn1-33Q and GST-Atxn1-86Q (Fig. 4c, upper panel). In the reciprocal experiment, GST-Atxn1-33Q and GST-Atxn1-86Q, but not GST, were pulled

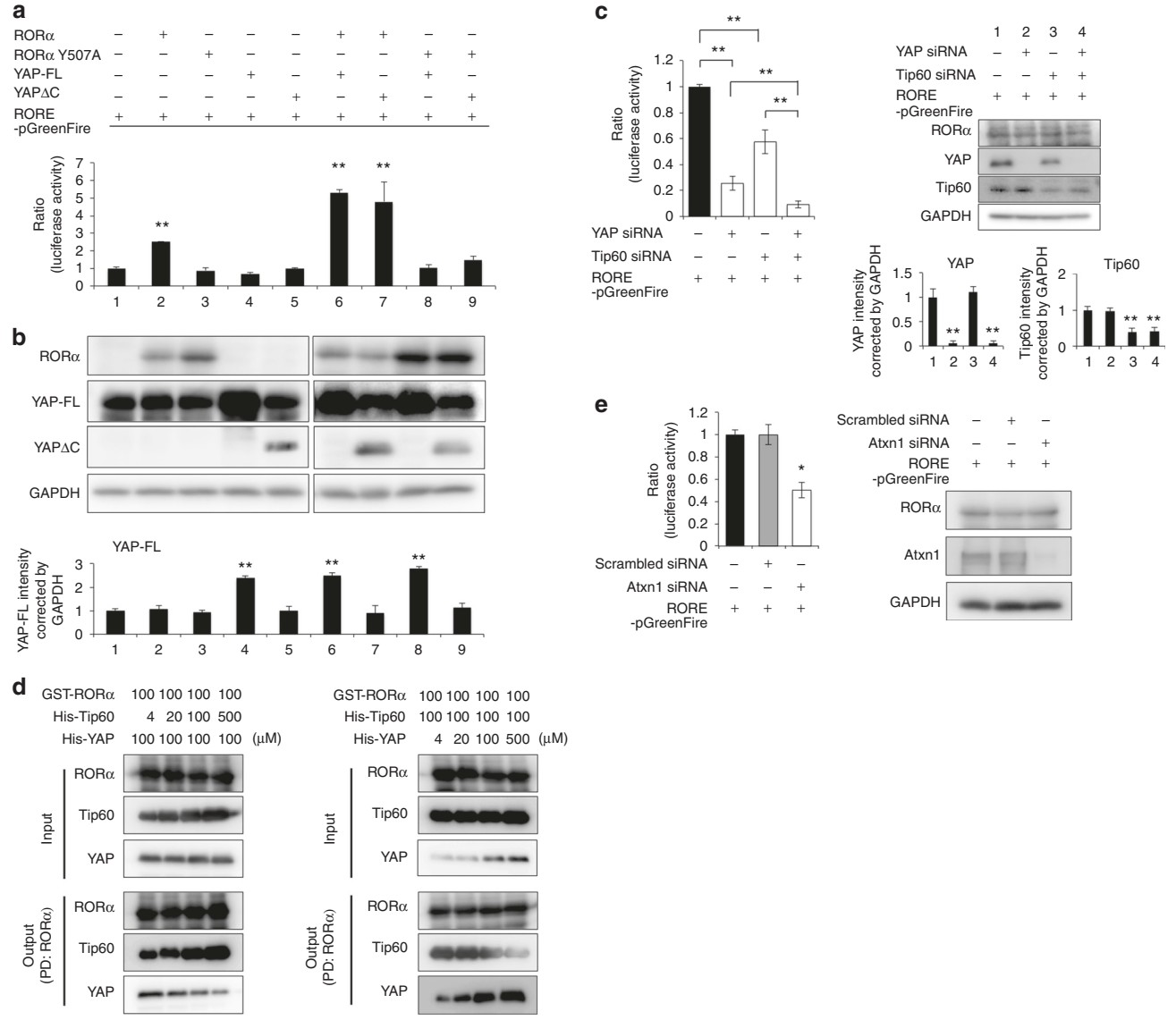

**Fig. 5** YAP/YAPdeltaC are co-factors of RORα. **a** Luciferase assays with COS-7 cells showing that RORα but not mutant RORα (RORα Y507A) lacking the consensus motif for the WW domain of YAP, acts with YAP or YAPdeltaC to transactivate luciferase gene expression via the RORα-responsive element. Single transfection of YAP or YAPdeltaC expression vector yielded insufficient transactivation. Double asterisks indicate statistical significance ($p < 0.01$, $N = 5$) in one-way ANOVA with post hoc Tukey's HSD test. **b** Protein expression levels of RORα, YAP, and YAPdeltaC in the luciferase assays shown in Fig. 5a, determined by western blot. Because YAP is expressed endogenously in COS-7 cells, the amount of YAP was quantified ($N = 5$). **c** Left: effect of YAP knockdown and/or Tip60 knockdown on RORα-mediated transcription, evaluated by luciferase assays in 293T cells expressing RORα. Asterisks indicate statistical significance ($p < 0.05$, $N = 8$) in one-way ANOVA with post hoc Tukey's HSD test. Right: levels of RORα, YAP, YAPdeltaC, and Tip60 are shown in lower panels. Double asterisks indicate statistical significance ($p < 0.01$, $N = 5$) in one-way ANOVA with post hoc Tukey's HSD test. **d** Pull-down assay showing that YAP and Tip60 compete for interaction with RORα. A GST fusion protein of a part of RORα (a.a. 424–523) was used in this assay, as described in Methods. **e** Effect of *Atxn1* knockdown on RORα-mediated transcription, evaluated by luciferase assays in 293T cells expressing RORα. Expression levels of RORα and Atxn1 are shown in right panel. Asterisks indicate statistical significance ($p < 0.05$, $N = 6$) in one-way ANOVA with post hoc Tukey's HSD test

down with His-YAP (Fig. 4c, lower panels). On the other hand, similar pull-down experiments did not reveal a direct interaction between GST-Atxn1 and RORα, as reported previously[12].

Because RORα co-precipitated with normal but not mutant Atxn1 (Fig. 4a, b), and a direct interaction between Atxn1 and RORα was not confirmed in a previous report[12], we hypothesized that the RORα–YAP–Atxn1 ternary complex forms via RORα–YAP and YAP–Atxn1 direct interactions. The ternary complex formed efficiently with normal Atxn1, but to a lesser extent with mutant Atxn1, suggesting that mutant Atxn1 might perturb integration of YAP into the ternary complex.

The similar experiments with 293T cells, which express RORα endogenously, at 2 days after transfection supported that normal and mutant Atxn1 equally interact with YAP/YAPdeltaC (Supplementary Fig. 1C, D). Again, RORα co-precipitated with normal but not mutant Atxn1, suggesting perturbation of YAP integration into the RORα–YAP–Atxn1 ternary complex by mutant Atxn1.

**RORα–YAP/YAPdeltaC–Atxn1 complex is functional.** Next, we investigated whether YAP/YAPdeltaC function as a

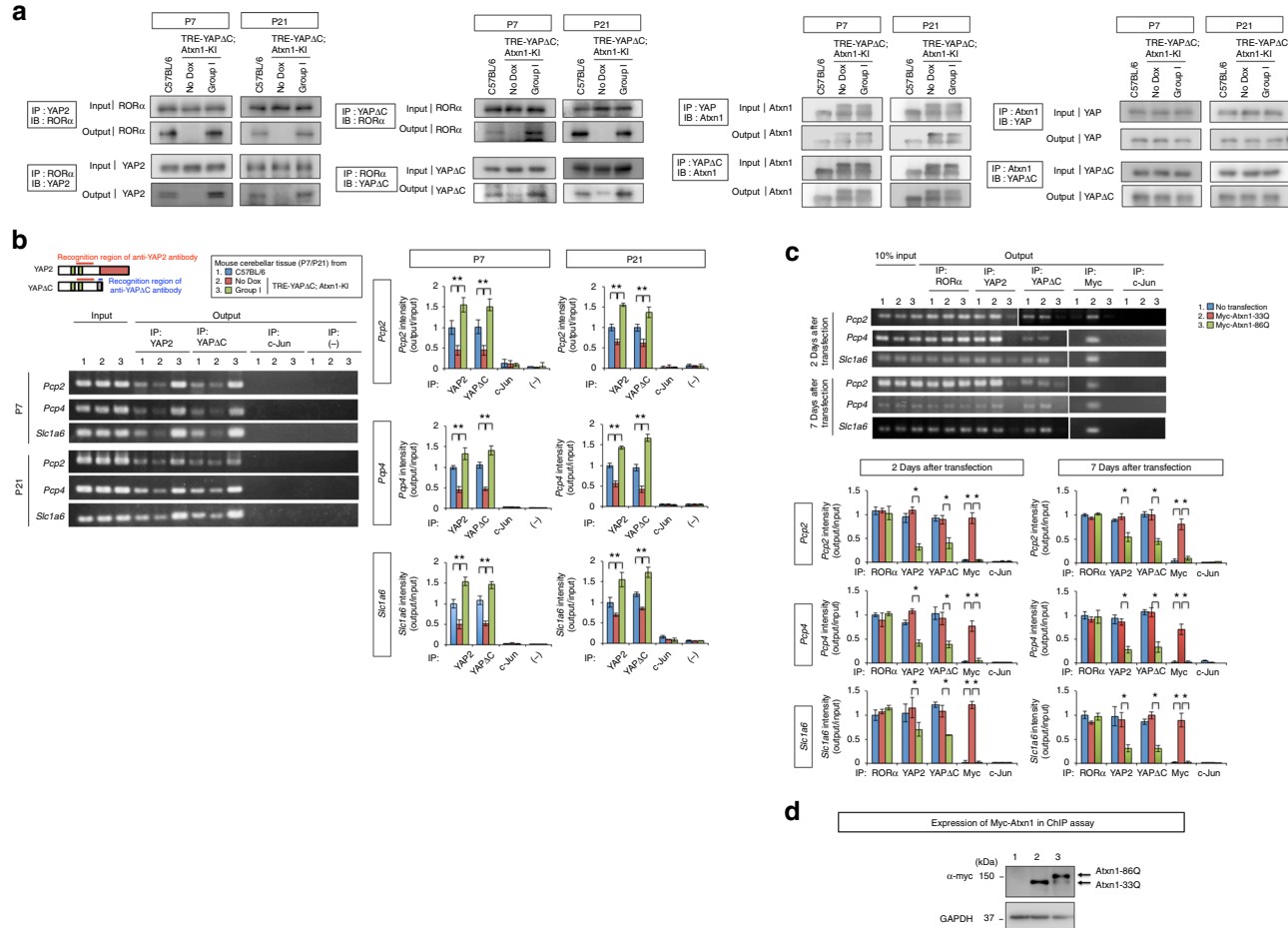

**Fig. 6** Mutant Atxn1 impairs the interaction of YAP/YAPdeltaC with RORα in vivo. **a** Immunoprecipitation analyses of mouse cerebellar cortex at P7 and P21 revealing reduced interaction between RORα with YAP/YAPdeltaC in Atxn1-KI mice (No-Dox) and the recovery of the interaction in Group I mice. **b** ChIP assay was performed on mouse cerebellar cortex using anti-YAP2 antibody (recognizing both full-length YAP and YAPdeltaC) or anti-YAPdeltaC antibody (recognizing only YAPdeltaC). The RORα-responsive elements in the upstream region of Pcp2, Pcp4, and Slc1a6 were amplified using specific primers. Quantitative analyses of band intensities are shown in the graphs at right. Double asterisks indicate statistical significance ($p < 0.01$, $N = 4$) in one-way ANOVA with post hoc Tukey's HSD test. **c** ChIP assay was performed on primary cerebellar neurons 2 or 7 days after transfection. Expression of Myc-Atxn1-86Q decreased the amount of YAP or YAPdeltaC in the RORα transcription complex on the cis-elements of Pcp2, Pcp4, and Slc1a6, whereas mutant Atxn1 did not change the amount of RORα attached to the cis-element. Immunoprecipitation of c-Jun was performed as a negative control. Double asterisks indicate statistical significance ($p < 0.01$, $N = 5$) in one-way ANOVA with post hoc Tukey's HSD test. **d** Protein levels of Myc-Atxn1-33Q and Myc-Atxn1-86Q, determined by western blot using anti-Myc antibody, in the samples shown in Fig. 6c

transcription co-activator for RORα (Fig. 5a, b). As mentioned above, COS-7 cells express a very low level of RORα but a high level of full-length YAP (Fig. 5b). Co-transfection of an effector plasmid expressing RORα (RORα-pCI neo), but not RORα-Y507A mutant lacking affinity for YAP/YAPdeltaC, increased luciferase activity from a RORα-responsive element on a reporter plasmid (RORE-pGreenFire) (Fig. 5a). In the absence of RORα, overexpression of YAP/YAPdeltaC did not affect luciferase activity (Fig. 5a). YAP/YAPdeltaC increased transcriptional activity in the presence of wild-type RORα but not of RORα-Y507A (Fig. 5a). Collectively, these results supported that YAP/YAPdeltaC function as transcriptional co-activators for RORα.

We knocked down YAP and/or Tip60 using siRNA in 293T cells which express RORα and YAP endogenously (Fig. 5c). YAP-siRNA dramatically decreased the level of YAP protein, and suppressed transcriptional activity proportional to the degree of knockdown (Fig. 5c). Tip60-siRNA decreased Tip60 protein level, albeit less efficiently, and also proportionally suppressed transcription (Fig. 5c), suggesting that YAP and Tip60 are co-factors

with equivalent activity. Double knockdown of YAP and Tip60 additively suppressed transcription from the RORα-responsive element (Fig. 5c), indicating that YAP and Tip60 have complementary functions.

In vitro pull-down assays revealed that YAP and Tip60 compete for binding to RORα. In these experiments, His-Tip60, His-YAP (full-length YAP2), and GST-RORα (a.a. 424–523, including the overlapping binding motifs) (Fig. 3a) were mixed in vitro, and GST-RORα was pulled down with glutathione–Sepharose. Increasing the amount of His-Tip60 decreased the amount of His-YAP pulled down with RORα, and vice versa (Fig. 5d).

We next investigated the role of Atxn1 in RORα-mediated transcription using 293T cells, in which Atxn1 is expressed endogenously (Fig. 5e). Atxn1-siRNA but not a scrambled siRNA markedly suppressed transcriptional activity (Fig. 5e). Collectively, these results indicated that the RORα–YAP/YAPdeltaC–Atxn1 complex is functional in the normal state, and that all components in the complex are essential for transcriptional activity.

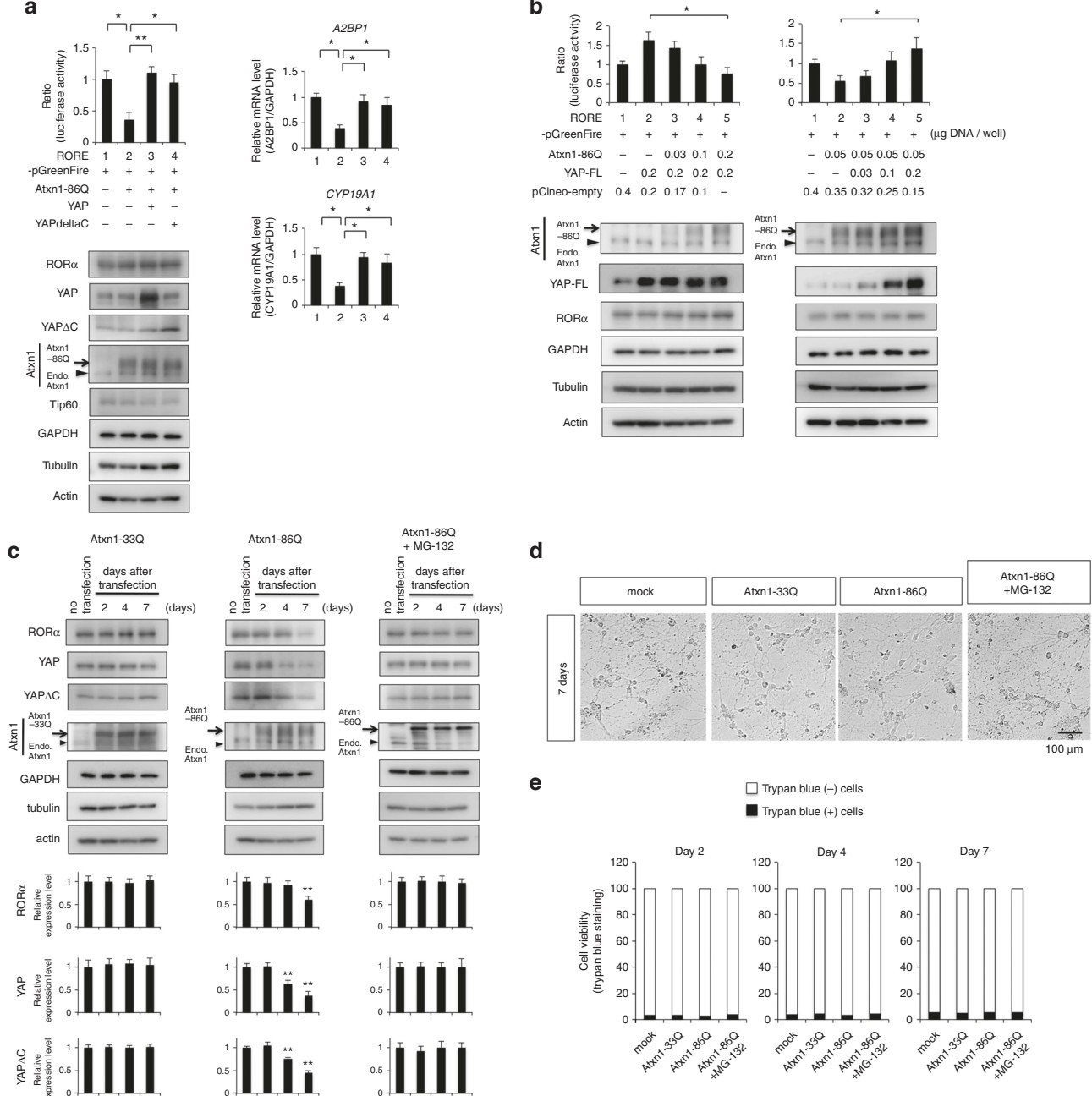

**Fig. 7** YAP/YAPdeltaC restores RORα target gene expression in vitro. **a** In primary cerebellar neurons, luciferase assays were performed 2 days after transfection of reporter and effector plasmids (left upper graph). Expression levels of RORα, Atxn1-86Q, YAP, YAPdeltaC, Tip60, GAPDH, α-tubulin, and β-actin at the time of the luciferase assay are shown in the lower panels. Atxn1-86Q suppressed RORα-mediated transcription immediately without decreasing the level of RORα or YAP. Transcriptional suppression was rescued by co-expression of YAP or YAPdeltaC (right graphs). Double asterisks indicate statistical significance ($p < 0.01$, $N = 7$) in one-way ANOVA with post-hoc Tukey's HSD test. **b** Dose-dependent inhibitory effect of Atxn1-86Q (left panels) and dose-dependent rescue effect of YAP2 (right panels) on RORα-mediated transcription. Double asterisks indicate statistical significance ($p < 0.01$, $N = 5$) in one-way ANOVA with post hoc Tukey's HSD test. Lower panels show the amounts of related proteins 2 days after transfection, as determined by western blot. **c** Protein levels of YAP, YAPdeltaC, Atxn1, GAPDH, α-tubulin, and β-actin were sequentially monitored until 7 days after transfection of Atxn1-33Q (left panels) or Atxn1-86Q (middle panels) to mouse primary neuron. YAP/YAPdeltaC were decreased from day 4, while RORα was decreased at day 7. The RORα level was not changed at 2 days, when RORα-mediated transcription was suppressed in luciferase assay. MG-132 treatment (0.3 μM, right panels) increased the level of YAP/YAPdeltaC and RORα decreased by Atxn1-86Q. Quantitative analyses are shown in the lower graphs. Double asterisks indicate statistical significance ($p < 0.01$, $N = 5$) in Dunnett's test. **d** Cell death was not markedly increased at day 7 after transfection of Atxn1-86Q. **e** Cell viability was examined at days 2, 4, and 7 after transfection by trypan blue dye exclusion test among non-transfected, Atxn1-33Q-transfected, Atxn1-86Q-transfected, and Atxn1-transfected + MG-132 primary cerebellar neurons. Black area represents mean value of % dye-positive neurons ($N = 5$)

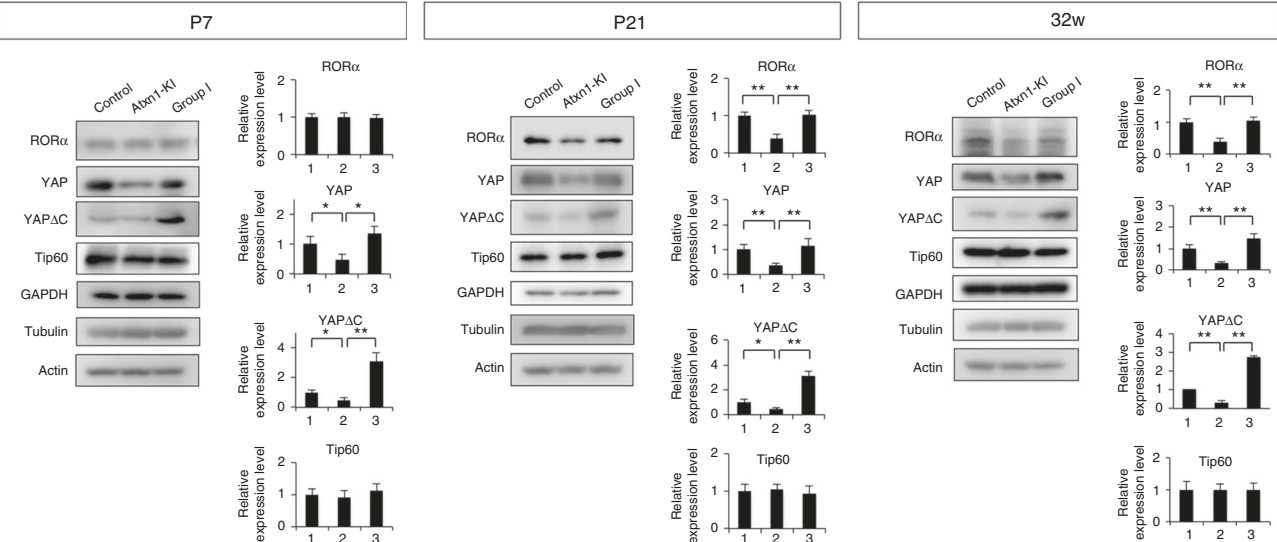

**Fig. 8** Expression of YAP, YAPdeltaC, and RORα in cerebellar cortex in vivo. Western blot analyses were performed with cerebellar cortex tissues to evaluate the protein levels of RORα, YAP, YAPdeltaC, Tip60, GAPDH, tubulin, or actin in three groups of mice at P7, P21, and 32 weeks. Band intensities were normalized against GAPDH, as summarized in the right graphs. One-way ANOVA with post hoc Tukey's HSD test was used for statistical analysis. **$p < 0.01$, $N = 4$

**Mutant Atxn1 depletes YAP/YAPdeltaC from the RORα complex**. We next asked how mutant Atxn1 affects interaction between YAP/YAPdeltaC and RORα in vivo. Using the cerebellar tissues prepared from mice at P7 and P21, we confirmed co-precipitation of full-length YAP (YAP2) and RORα (Fig. 6a, left panels). RORα co-precipitated with YAP was decreased in Atxn1-KI mice (No Dox) than C57BL/6 mice, but recovered in Group I mice (Fig. 6a, left upper panels). In the reciprocal precipitation of RORα, co-precipitated YAP was reduced in Atxn1-KI mice but recovered in Group I mice (Fig. 6a, left lower panels). Similarly, the in vivo interaction between YAPdeltaC and RORα was attenuated in Atxn1-KI mice (No Dox), but recovered in Group I mice (Fig. 6a, right middle left panel). Normal and mutant Atxn1 similarly interacted with YAP/YAPdeltaC (Fig. 6a right and middle right panels).

To investigate the in vivo protein–protein interaction between YAP/YAPdeltaC and RORα on cis-elements in the RORα target genes[20] (Pcp2, Pcp4, or Slc1a6) in three groups of mice (C57BL/6, No-Dox, and Group 1) at P7 and P21 (Fig. 6b), we performed ChIP assays with anti-YAP2 or anti-YAPdeltaC-specific antibody (the latter targets the unique sequence of YAPdeltaC) and PCR-amplified the RORα-responsive elements in the upstream regions of the target genes (Fig. 6b, left panels). In all three genes, the amount of amplified RORα-responsive element was lower in Atxn1-KI mice (No Dox) than in background C57BL/6 mice, but recovered in Group I mice (Fig. 6b, left panels and right graphs). The RORα-responsive element was not amplified from a sample precipitated with anti-c-Jun antibody (negative control) (Fig. 6b). Collectively, the results of the immunoprecipitation assays (Fig. 6a) and ChIP assays (Fig. 6b) suggested that mutant Atxn1 deprived YAP/YAPdeltaC from the RORα transcription complex in vivo.

Next, we investigated in chronological order when mutant Atxn1 deprived YAP/YAPdeltaC from the RORα transcription complex on target genes (Fig. 6c). We transiently expressed Myc-tagged normal/mutant Atxn1 in primary cerebellar neurons, which endogenously express RORα, YAP/YAPdeltaC, and normal Atxn1, and subjected these cells to ChIP assays. Already at 2 days after transfection, YAP/YAPdeltaC were not present in the RORα transcription complex on RORα-responsive element (Fig. 6c), and the deprivation of YAP/YAPdeltaC continued to day 7 (Fig. 6c). Although RORα was invariably bound to the

response element in its target genes, only Myc-Atxn1-33Q, but not Myc-Atxn1-86Q, was present in the RORα transcription complex on the cis-element (Fig. 6c). Since Myc-Atxn1-33Q and Myc-Atxn1-86Q were equivalently expressed in primary neurons (Fig. 6d), the results indicated that Myc-Atxn1-86Q could not efficiently approach to the RORα-responsive element.

We also performed ChIP assays in 293T cells. The results confirmed that Myc-Atxn1-33Q, but not Myc-Atxn1-86Q as well as bound YAP/YAPdeltaC, was present in the RORα transcription complex located on the cis-element of the three target genes (Supplementary Fig. 2A, B). Again, expression levels of Myc-Atxn1-33Q and Myc-Atxn1-86Q were equivalent in these samples (Supplementary Fig. 2B).

**YAPdeltaC recovers RORα-transcription**. To test the effects of mutant Atxn1 on RORα-dependent transcription, we performed luciferase assays in primary cerebellar neurons at 2 days after transfection. As expected, mutant Atxn1 suppressed transcription from RORα-responsive promoter, and YAP/YAPdeltaC rescued the suppression (Fig. 7a). The suppression by mutant Atxn1 and the rescue by YAP/YAPdeltaC were dose-dependent (Fig. 7b). In addition, we found that YAP/YAPdeltaC levels decreased from 4 days after transfection of mutant Atxn1 and RORα level followed from day 7 (Fig. 7c). Proteasome inhibitor MG-132 blocked the reduction of RORα and YAP/YAPdeltaC levels (Fig. 7c, right panels), suggesting that non-functional RORα and YAP/YAPdeltaC were degraded by the ubiquitin–proteasome system. Morphological analysis (Fig. 7d) and cell death assay with trypan blue staining (Fig. 7e) excluded the concern that these changes were due to the unhealthy state and/or cell death of primary cerebellar neurons by expression of mutant Atxn1.

Luciferase assays with 293T cells confirmed that YAP/YAPdeltaC rescued mutant Atxn1-induced suppression of RORα-dependent transcription (Supplementary Fig. 3A), that suppression and recovery were dose-dependent (Supplementary Fig. 3B), that mutant Atxn1 could suppress RORα-mediated transcription even without reduction of RORα (Supplementary Fig. 3A, B), and that mutant Atxn1 decreased YAP/YAPdeltaC and RORα sequentially (Supplementary Fig. 3C). The proteasome inhibitor MG-132 blocked the reduction in RORα and YAP/

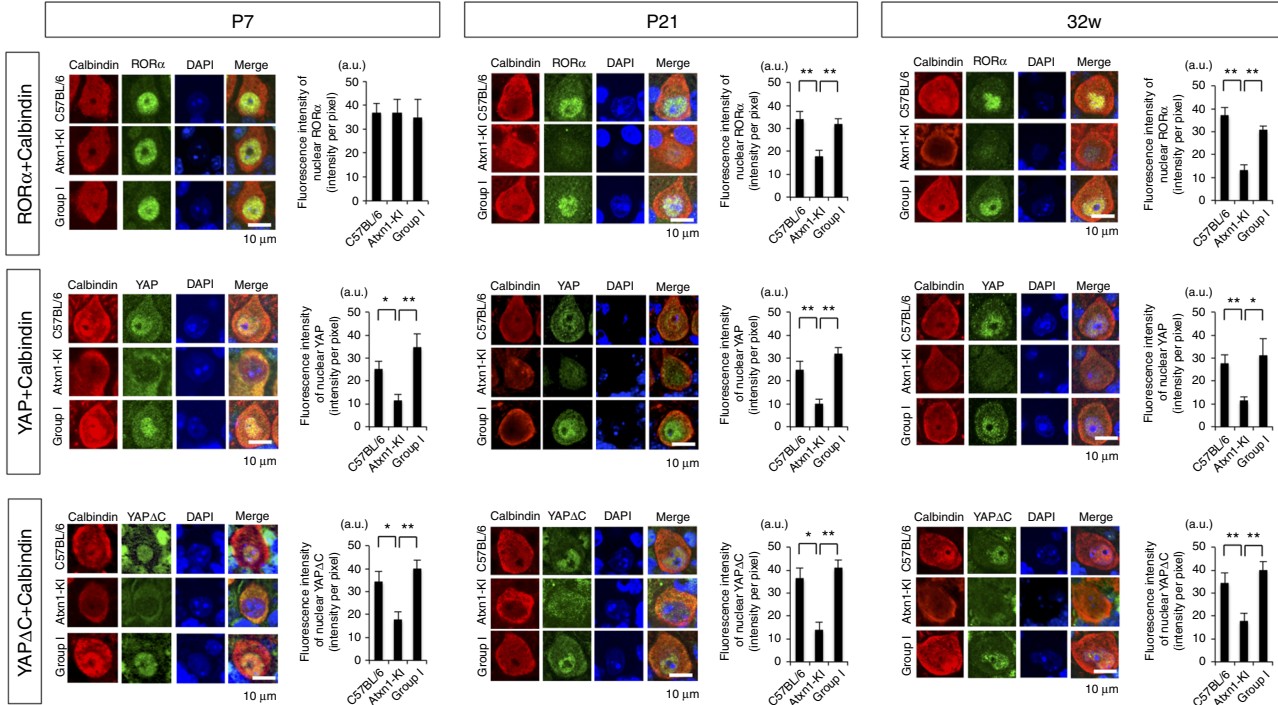

**Fig. 9** Expression of YAP, YAPdeltaC, and RORα in Purkinje cells in vivo. Purkinje cells were co-stained for Calbindin, a Purkinje cell-specific marker, and RORα, YAP, or YAPdeltaC in three groups of mice at P7, P21, and 32w. Graphs at right show quantitative analyses of RORα, YAP, or YAPdeltaC signals in nuclei of Purkinje cells. Signal intensities were acquired from more than 100 Purkinje cells, randomly selected from 30 slides, with six mice in each group. One-way ANOVA with post hoc Tukey's HSD test was used for statistical analysis. **$p < 0.01$

YAPdeltaC levels also in 293T cells (Supplementary Fig. 3C). Based on a phase-contrast image of the transfected cells, we can exclude that Atxn1-induced cell death affected transcriptional activity in the assay (Supplementary Fig. 3D).

Based on these results, we hypothesize that in vivo, mutant Atxn1 disturbs RORα-mediated transcription over short time-scales by directly interacting with YAP/YAPdeltaC and depleting these co-activators from the transcriptional complex. Meanwhile, over longer periods, mutant Atxn1 accelerates degradation of YAP/YAPdeltaC and RORα, resulting in a reduction in their protein levels (Supplementary Fig. 4).

**YAP/YAPdeltaC, RORα, and Atxn1 in Purkinje cells in vivo**. To test the hypothesis that mutant Atxn1 in *Atxn1*-KI mice depletes YAP/YAPdeltaC from the transcription complex, thereby RORα-mediated transcription during development, we next sought to verify that all of the relevant factors are actually co-expressed in Purkinje cells, and examined how expression of mutant Atxn1 affects RORα and YAP/YAPdeltaC in Purkinje cells in vivo. For this purpose, we performed western blot of C57BL/6, No Dox, and Group I mice at P7, P21, and 32 weeks (Fig. 8).

RORα signal intensity was not changed at P7 but decreased in *Atxn1*-KI mice at P21 and 32 weeks (Fig. 8). In this regard, our results in mutant *Atxn1*-KI mice (Sca1[154Q/2Q]) were consistent with those of a previous study reporting reduced expression of RORα at P35 (5 weeks)[12] in Pcp2/L7 promoter-driven Tet-Off Atxn1-82Q transgenic mice[21] and of a ROR-target gene EAAT4 at P21 (3 weeks) in B05 Pcp2/L7 promoter-driven Atxn1-82Q simple transgenic mice[22, 23]. Meanwhile, the decreases in YAP and YAPdeltaC levels in Atxn1-KI mice started at P7 and preceded the change in RORα at P21 (Fig. 8). All changes in RORα YAP, and YAPdeltaC levels were rescued by over-expression of YAPdeltaC in Group I mice from P7 to 32 weeks

(Fig. 8). By contrast, the Tip60 level was not reduced in *Atxn1*-KI mice at any time point (Fig. 8).

Next, we performed immunohistochemistry analyses of YAP/YAPdeltaC, RORα, or Tip60 with Calbindin in C57BL/6, No Dox, and Group I mice at P7, P21, and 32 weeks (Fig. 9). In Purkinje cells of background mice, all four proteins were expressed at all time points (Fig. 9). Signal intensities of YAP and YAPdeltaC were reduced in Purkinje cells of *Atxn1*-KI from P7 to 32 weeks. Meanwhile RORα was decreased from P21. The discrepancy between YAP/YAPdeltaC and RORα at P7 was consistent with the result in western blot. All changes in RORα YAP and YAPdeltaC levels were rescued again by YAPdeltaC in Group I mice from P7 to 32 weeks (Fig. 9).

We checked the amount and intracellular distribution of Atxn1 using anti-Atxn1 antibody 11NQ, along with YAP, YAPdeltaC, RORα, or Tip60, at P21 (Supplementary Fig. 5) and 32 weeks (Supplementary Fig. 6). Expression levels of Atxn1 in the nucleus were not different significantly among three groups. Although the symptoms have already appeared by this age, nuclear inclusion bodies are not detected before 40 weeks[16].

Collectively, immunohistochemistry revealed that all relevant factors are expressed in the most vulnerable Purkinje cells, and that not only RORα but also YAP/YAPdeltaC levels were reduced by mutant Atxn1 in vivo. Thus, the results supported our hypothesis based on the chronological relationship of these four molecules.

**YAPdeltaC rescues RORα target genes in KI mice**. Next, we investigated how RORα-dependent transcription was altered in the three groups of mice. To this end, we performed RT-qPCR analysis to evaluate expression levels of two RORα target genes[24], *A2BP1* and *CYP19A1*, using Purkinje cells punched out from frozen sections of mouse cerebellum, as described previously[25] (Fig. 10). We selected these two target genes for qPCR (Fig. 10a),

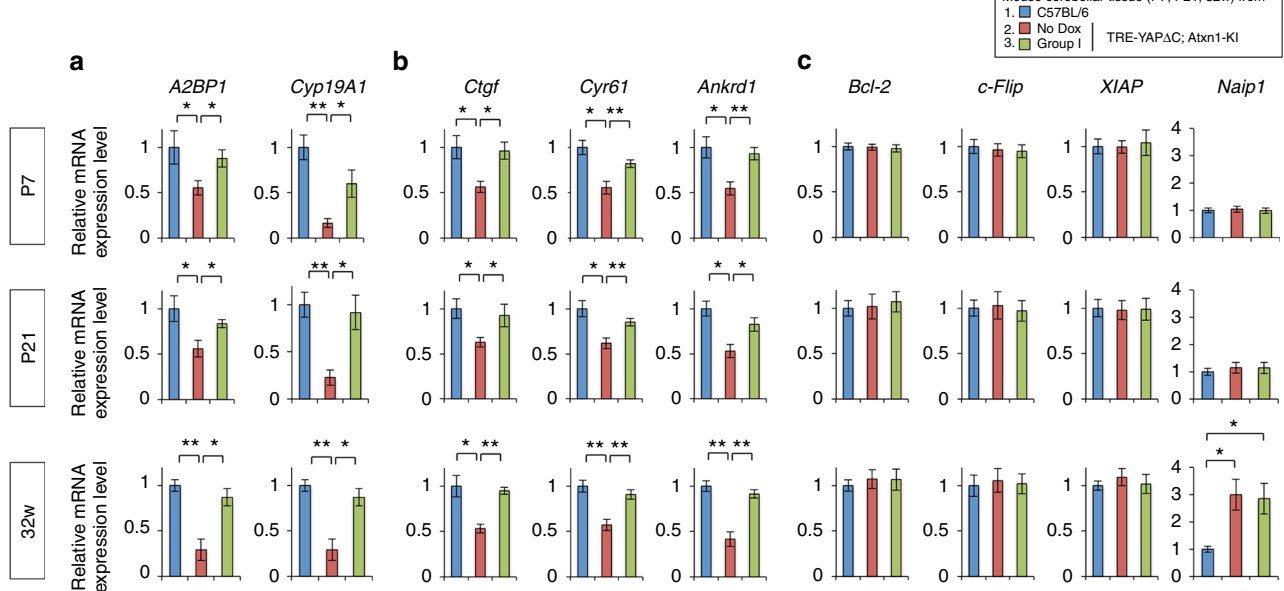

**Fig. 10** YAPdeltaC restores RORα target gene expression in Group I mice in vivo. **a** RT-qPCR confirmed that YAPdeltaC overexpression in Group I mice rescued the reduced expression of two RORα target genes (*A2bp1* and *Cyp19a1*) in Purkinje cells of *Atxn1*-KI mice at P7, P21, and 32 weeks. **b** RT-qPCR revealed a similar change in YAP target genes independent of RORα: *Cyp19a1* (TEAD-dependent[26, 27]), *Cyr61* (TEAD-dependent[26, 27]), and *Ankrd1* (TEAD-dependent[26, 27]). **c** RT-qPCR detected no remarkable changes in expression of representative anti-apoptotic genes (Bcl-2, c-Flip, XIAP, Naip1) in Purkinje cells in three types of mice. Single and double asterisks indicate statistical significance ($p < 0.05$ and $p < 0.01$, $N = 3$) in one-way ANOVA with post hoc Tukey's HSD test

in addition to the target genes[20] used for the ChIP assays (*Pcp2*, *Pcp4*, and *Slc1a6*) (Fig. 6b, c) in order to generalize the effect of YAPdeltaC on RORα target genes. As expected, expression levels of the additional genes were reduced in Purkinje cells of *Atxn1*-KI mice, but recovered in Group I mice (Fig. 10a). The decrease and recovery in expression levels of RORα target genes at P7 (Fig. 10a), when levels of the co-activators YAP/YAPdeltaC but not the transcription factor RORα itself were reduced (Figs. 8 and 9), were consistent with the reduced transcriptional activity in the luciferase assay with primary cerebellar neurons at day 2 after transfection, when YAP/YAPdeltaC but not RORα levels were reduced (Fig. 7). RT-qPCR analysis also revealed that expression of multiple YAP target genes[26, 27] was similarly reduced and recovered in Purkinje cells of three types of mice (Fig. 10b).

YAP overexpression might make neurons more resistant to mutant Atxn1-induced cell death, dependently on the upregulation of anti-apoptotic genes in developing cerebellar neurons. However, expression levels of representative anti-apoptotic genes such as Bcl-2, c-Flip, and XIAP were not altered (Fig. 10c). Regarding Naip1, which was detected in our previous study as an increased gene during TRIAD[13], was not changed at P7 and P21 (Fig. 10c). At 32 weeks, Naip1 was already increased in *Atxn1*-KI mice similarly to the condition of TRIAD[13] (Fig. 10c). These results did not support that apoptosis is responsible for the decrease of Purkinje cell death in *Atxn1*-KI mice, while it remained possible that other types of cell death contribute to the pathological process from P21 to 8 weeks of age. These results supported our hypothesis that supplementation with YAPdeltaC ameliorates Atxn1-induced pathology in vivo via recovery of RORα-dependent transcription.

Furthermore, we tested whether YAP or YAPdeltaC induced de-differentiation of primary cerebellar neurons prepared at P7 (Supplementary Fig. 7). The concern was basically excluded, since YAP or YAPdeltaC even at high expression levels (Supplementary Fig. 7A) did not induce tumor-like foci (Supplementary Fig. 7B) or did not increase the number of cells positive for Ki67 (a tumor

marker) and Sox2 (a de-differentiation marker), as revealed by immunohistochemistry (Supplementary Fig. 7C) or western blot analysis (Supplementary Fig. 7D).

**YAPdeltaC rescues DNA damage in KI mice**. YAP is an essential component of the Hippo signaling pathway, which influences DNA repair and apoptosis in cancer cells[28, 29], YAP/YAPdeltaC protects against TRIAD[13]. Accordingly, we investigated whether supplementation with YAPdeltaC would rescue DNA damage in Purkinje cells of *Atxn1*-KI mice.

Consequently, we performed immunohistochemistry and western blot analyses to monitor the increase of DNA damage markers, such as γH2AX and 53BP1, in cerebellar tissues of *Atxn1*-KI mice at P21 (Supplementary Fig. 8A, B). Immunohistochemistry revealed that DNA damage was elevated, especially in Purkinje cells, and this increase in DNA damage was rescued by in vivo co-expression of YAPdeltaC in Group I mice (Supplementary Fig. 8A, B, upper panels). Western blot analyses of cerebellar tissues supported this conclusion (Supplementary Fig. 8A, B, lower panels). We also confirmed by western blot that the level of RpA1, a critical molecule for multiple forms of DNA repair involved in SCA1 pathology[25, 30], was elevated in Purkinje cells, probably in response to the elevated DNA damage in Atxn1-KI mice, but was restored to normal by co-expression of YAPdeltaC in vivo (Supplementary Fig. 8C).

**Discussion**

The mouse genetic analyses performed in this study revealed that expression of YAPdeltaC during development, but not in adulthood, markedly ameliorated the behavioral and pathological phenotypes of *Atxn1*-KI mice in adulthood (Figs. 1 and 2). Our results suggest that certain molecular signatures generated during development significantly influence late-onset phenotypes, and that these signatures act by restoring gene expression to optimal levels. YAP/YAPdeltaC and RORα are candidate components of

such a molecular signature. At short timescales, mutant Atxn1 binds to YAP/YAPdeltaC and disrupts RORα-mediated transcription by depleting YAP/YAPdeltaC from the RORα complex. Shuttling of nuclear proteins between nuclear bodies, where transcription-related factors are stored, and the nucleoplasm, where the factors assemble to on histone-free and relaxed regulatory elements of target genes, is impaired by mutant Atxn1 and other polyQ proteins[31–33]. Thus, mutant Atxn1 could slow down the dynamic interactions of direct binding partners such as YAP/YAPdeltaC, and thereby impair transcription.

Over longer periods of more than 7 days, the interaction with mutant Atxn1 decreased YAP/YAPdeltaC and RORα levels in a cell culture model via accelerated proteasome-mediated degradation (Fig. 7c). A similar decrease in YAP/YAPdeltaC and RORα levels was confirmed in vivo in mutant *Atxn1*-KI mice at 21 days and 32 weeks of age (Fig. 8, Supplementary Fig. 5 and 6). Therefore, both acute and chronic mechanisms are likely to contribute to the suppression of RORα-mediated transcription of critical target genes that eventually causes the late-onset pathology of SCA1.

We employed the NSE promoter for our Tet-ON system. If this promoter drove YAPdeltaC in cells other than Purkinje cells, this expression might have affected RORα-mediated transcription or TRIAD and contributed to the lifespan elongation observed in Group I mice. To explore this possibility, it will be necessary to characterize the gene expression patterns of NSE in other cell types, other regions of the brain, and other organs. Based on publicly available information in databases such as GeneCards (http://www.genecards.org/cgi-bin/carddisp.pl?gene=ENO2) and MGI (http://www.informatics.jax.org/gxd#gxd=nomenclature%3DNSE%26vocabTerm%3D%26annotationId%3D%26locations%3D%26locationUnit%3Dbp%26structure%3D%26structureID%3D%26theilerStage%3D0%26results%3D100%26startIndex%3D0%26sort%3D%26dir%3Dasc%26tab%3Dresultstab), expression levels of NSE are very low in organs other than the brain, and within the brain expression is restricted to neurons. Based on our previous results, obtained using transgenic mice in which the same NSE promoter drove HMGB1[25], we can speculate about the gene's expression pattern. Although expression was detected in neurons of various brain regions, the levels in Purkinje cells were substantially higher than in other neurons in different regions[25]. We also measured NSE expression in other organs, and confirmed that the levels were very low (data not shown).

Therefore, although the effects of YAPdeltaC on RORα-mediated transcription or TRIAD (discussed later) in other brain regions or organs might have additively contributed to the phenotypic recovery of Group I mice, we consider it likely that a large proportion of the rescue effect of YAPdeltaC is derived from its activity in Purkinje cells. This hypothesis effectively explains why our results in *Atxn1*-KI mice (Sca1[154Q/2Q]) were consistent, but slightly improved with respect to lifespan with previous results obtained in transgenic mice harboring Pcp2/L7-promoter-driven Atxn1-82Q[21], in which mutant Atxn1 expression could be switched on or off specifically in Purkinje cells[12]. This speculation is also consistent with recent observations of polyQ/CAG repeat disease pathology in non-neuronal cells[11].

Moreover, in human patients, most cases progress motor ataxia due to the loss of Purkinje cells, become bed-ridden, suffer insufficient food intake and aspiration pneumonia due to ataxia of pharyngeal muscles, and die. SCA1 patients with severer loss of Purkinje cells die at younger ages than SCA6 patients in which loss of Pukinje cells are not prominent. Therefore, the lifespan shortening/recovery in our mouse models based on Purkinje cell pathology may mimic the human natural history better than previous experimental reports.

Our findings revealed that YAP/YAPdeltaC interacts with RORα via the second WW domain (Fig. 3). Interestingly, YAP/YAPdeltaC also use the second WW domain to interact with mutant Htt[15]. In both cases, YAP/YAPdeltaC are deprived of their partner transcription factors. In HD pathology, transcription mediated by TEAD, a partner transcription factor of YAP/YAPdeltaC, is impaired[15]. This study provides the first evidence that RORα is a partner transcription factor of YAP/YAPdeltaC, and that RORα-mediated transcription is impaired in SCA1 pathology. Together, these two studies imply that YAP/YAPdeltaC-related gene expression represents a common pathological mechanism shared by multiple polyQ diseases.

It is possible that YAP/YAPdeltaC influence apoptosis and increase resistance against SCA1-associated cerebellar degeneration, including the reduction in Purkinje cell numbers, independently of the hypothesized effect of YAP/YAPdeltaC on RORα-mediated transcription. However, YAP/YAPdeltaC did not affect expression levels of anti-apoptotic genes in Purkinje cells from P7 to 32 weeks (Fig. 10). Since previous data suggested that cell death of Purkinje cells occur from 16 to 32 weeks in *Atxn1*-KI mice[16, 25, 34–36], these data do not support the idea that the late-onset effects of YAPdeltaC in Group I mice are due to suppression of apoptosis.

Another important observation was the delay in cell death downstream of YAP/YAPdeltaC dysfunction. Recently, we reported that a new form of necrosis, dependent on Hippo pathway signaling, contributes to the pathology of Huntington's disease[15]. The impairment of TEAD–YAP-mediated transcription leads to a lengthy process of necrotic cell death. The morphological and molecular features of TEAD–YAP-dependent cell death were very similar to those of TRIAD[15], a Type 3 necrotic cell death induced by repression of transcription with the RNA polymerase II inhibitor α-amanitin[13]. Accordingly, these two forms of cell death are considered identical. Ultra-structural and biochemical analyses of human postmortem tissue revealed that TRIAD actually occurs in human HD brains[14]. Given that target genes of TEAD-YAP transcription were suppressed in *Atxn1*-KI mice and recovered in Group I mice (Fig. 10), TEAD-YAP-mediated TRIAD might be a possible explanation for the late-onset effect of YAPdeltaC in Group I mice.

From the perspective of neuronal dysfunction, dendrite abnormalities were observed in *Atxn1*-KI mice at 13 weeks, an age when the cell death of Purkinje cells was not obvious[34]. Interestingly, the Hippo pathway has recently been implicated in synapse formation in *Drosophila*[37]. These observations suggest that functional impairment of YAP/YAPdeltaC during development until 8 weeks of age induces sequential processes, ranging from neuronal dysfunction to cell death in adulthood. Given that RORα plays a critical role in SCA1-associated neurodegeneration[12], and considering that the results our study reveal the close relationship between YAP/YAPdeltaC and RORα, it is very plausible that YAP/YAPdeltaC contribute to the adult-onset functional pathology of SCA1 by impairing RORα-mediated transcription during development. However, the effect exerted by YAP/YAPdeltaC on SCA1 pathology via TEAD-mediated transcription should be investigated in future studies.

During the process of our paper evaluation, another group reported that TAZ, the mammalian homolog of YAP, functioned as a co-activator of ROR to promote Th17 cell differentiation through similar biochemical mechanism[38]. These two independent but consistent results might open a new pathway from Hippo pathway to ROR family receptors that could be applied to multiple organs.

Our results revealed that mutant, but not wild-type, Atxn1 depletes YAP/YAPdeltaC from the RORα complex. The first mechanism underlying the depletion would be proteasome-

dependent degradation of YAP/YAPdeltaC after the YAP/YAPdeltaC–mutant Atxn1 complex formation (Fig. 7c, Supplementary Fig. 3C). Second, presumably the nuclear dynamics of YAP/YAPdeltaC would be impaired by interaction with mutant Atxn1, as with RpA1 and VCP, whose nuclear dynamics are slowed and diminished by interaction with the mutant protein[30], [39]. In addition, the nuclear sites of interaction between YAP/YAPdeltaC and Atxn1 and between RORα and YAP/YAPdeltaC are likely to differ, as demonstrated previously for different combinations of transcription factors that ultimately execute integrated functions[40]. Therefore, interaction of mutant Atxn1 with YAP/YAPdeltaC in one subdomain would prevent interaction of YAP/YAPdeltaC with RORα in another. However, the second explanation remains a hypothesis that should be critically tested in future analyses.

The roles of YAP in DNA repair remain controversial. Two previous studies reported that YAP induces apoptosis in DNA-damaged cancer cells[28, 29]. This function is mediated by the YAP–p73 complex, which promotes expression of pro-apoptotic genes[41]. Meanwhile, in non-dividing neurons, YAP/YAPdeltaC functions as a protective factor against transcriptional repression-induced necrosis dependent on the YAP–TEAD complex[13, 15]. In that context, DNA damage is promoted by transcriptional inhibition[42], whereas YAP/YAPdeltaC decreases the level of DNA damage (Supplementary Fig. 8). Conversely, the discrepant functions of YAP/YAPdeltaC in DNA repair suggest that the proteins' primary roles are related to transcriptional regulation, and that any effects on DNA damage are secondary.

Regardless of the causative pathway, elevated DNA damage in stem cells decreases production of neurons. However, we did not observe a reduction in the abundance Purkinje cells at P21 (Fig. 2a). Instead, our analysis revealed that developmental expression of YAP/YAPdeltaC before 8 weeks (Fig. 1a) is protective against late-onset death of Purkinje cells after 16 weeks[16, 25, 34–36] (Fig. 2b). Atxn1 expression begins at E13.5–15 (http://www.informatics.jax.org/marker/MGI:104783), and markedly increases at P14[43]. In addition, protein aggregation and RNA foci of Atxn1 was not detected in Atxn1-KI mice until 40 weeks[16] far later than the onset age of motor symptom. Collectively, these findings suggest that the key mechanism by which YAPdeltaC ameliorates delayed cell death would be normalization of RORα-dependent and/or TEAD-dependent genes between E13.5 and 8 weeks (Fig. 1a). Therefore, identification of these downstream target genes would enable us to manipulate the late-onset cell death of SCA1 in the future.

## Methods

**Ethics**. This study was performed in strict compliance with the recommendations in the Guide for the Care and Use of Laboratory Animals of the National Institutes of Health (USA). This study was also approved by the Committees on Gene Recombination Experiments, Human Ethics, and Animal Experiments of the Tokyo Medical and Dental University (2010-215C13, 2014-5-4, and 0160328A, respectively).

**Animals**. The mutant Atxn1-KI mouse (Sca1[154Q/2Q] mouse) was a generous gift from Professor Huda Y. Zoghbi (Baylor College of Medicine, TX, USA)[16]. The original background of the Sca1[154Q/2Q] mouse was B6.129S, but subsequently the Zoghbi group backcrossed the strain with C57BL/6. Atxn1-154Q male mice were further crossed with C57BL/6 female mice more than 10 times in our laboratory, and the phenotypes of the resultant Sca1[154Q/2Q] mice in the B6 background were described previously[25]. Next, conditional and inducible YAPdeltaC-ins61 transgenic mice were generated using the Tet-on system. Briefly, reverse tetracycline-controlled transactivator (rtTA) transgenic mice under the control of the rat NSE (neuron-specific enolase) promoter (NSE-rtTA) were crossed with TRE (tetracycline-responsive element) promoter-regulated YAPdeltaC-ins61 transgenic mice. Double-heterozygous transgenic mice were crossed to generate double-homozygous transgenic mice, and the resultant female double-homozygous transgenic mice (NSE-rtTA/YAPdeltaC) were crossed with male Sca1[154Q/2Q] mice in the B6 background to generate triple-transgenic mice (rtTA/YAPdeltaC/Atxn1-KI mice). Thus, the genetic backgrounds of YAPdeltaC-Tg, NSE-rtTA, NSE-rtTA/YAPdeltaC, and NSE-rtTA/YAPdeltaC/Atxn1-KI mice were all C57BL/6.

**Rotarod test**. Male 5–21-week-old control (C57BL/6J mice), Atxn1-KI, and Atxn1-KI/YAPΔC mice were tested. Mice were placed on a rotating rod (shaft diameter: 3.2 cm, lane width: 5.7 cm, fall height: 16.5 cm. Five Station Rota-Rod Standalone for Mouse, ENV-577M, MED Associates Inc®, USA) in accelerating speed mode 8 (i.e., rotation speed was linearly increased from 3.5 to 35 r.p.m. over 300 s and then maintained at 35 r.p.m. for additional 600 s). The amount of time spent on the drum until the mouse fell off was measured four times per day over 3 days, with 10-min rest intervals between the measurements.

**Plasmid construction**. Human and rat YAP2 (full-length YAP) and rat YAPdeltaC (YAPdeltaC-ins61) cDNAs were generated previously[13]. The cDNAs were subcloned into pCIneo (Promega, Madison, WI, USA) containing the FLAG tag (FLAG-YAP-pCIneo and FLAG-YAPdeltaC-pCIneo). Plasmids with mutations in the first and/or second WW domain mutations were purchased from Addgene (2xFlag CMV2-YAP2-1st and/or 2nd WW mutation, MA, USA). To construct human RORα-pCIneo, human RORα cDNA (1591 bp, nt 86-1657 of NM_136241) was amplified from total RNA of 293T cells using primers 5′-CCGGAATT-CATGGAGTCAGCTCCGGCA-3′, and 5′-ACGCGTCGACTTACCCAT-CAATTTGCATTGCTGGC-3′. After digestion with EcoRI or SalI, the RORα cDNA was subcloned into pCI-neo (Promega). The Y507A mutation (RORα-Y507A-pCIneo) was generated using primers 5′-CATTTTCCTCCATTAGC-CAAG-3′, and 5′-GTGAACAACTCCTTGGCTAAT-3′. To construct human RORE (RORα-responsive element)-pGreenfire, the upstream genomic fragment of human NR1D1 gene containing RORα response element (1844 bp, nuclear receptor subfamily 1 group D member 1, nt 632–2476 of NM_021724) was amplified from human genomic DNA (#6550-1, CLONTECH, CA, USA) using primers 5′-CCGGAATTCAAAGGGGGTCACATTTCCTTTCC-3′ and 5′-CTAGTCTAGATGCCCCAGTGACACACTTTT-3′, and subcloned into pGreen-Fire1 (#TR011PA-1, System Biosciences, CA, USA) after digestion with EcoRI or XbaI. Myc-Atxn1-0Q, Myc-Atxn1-33Q, and Myc-Atxn1-86Q were generated as described previously[44].

**Cerebellar neuron culture and plasmid transfection**. Cerebellar neurons were prepared from C57BL/6J mice at postnatal day 7. Briefly, dissected cerebella were digested at 37 °C with 0.05% trypsin (GIBCO) for 15 min. Tissue was passed through a 70-μm cell strainer (BD). Cells were cultured in DMEM (Sigma) containing 25 mM D-glucose, 4 mM L-glutamine, 25 mM KCl, and 10% fetal bovine serum in the presence of 2 μM AraC. Plasmid transfections were performed using Viromer Red (OriGene).

**Western blot analysis**. Mouse cerebellum tissues were dissolved in lysis buffer (10 mM Tris-HCl (pH 7.5), 150 mM NaCl, 1 mM EDTA, 1% NP-40) with protease inhibitor cocktail (#539134, 1:100 dilution, Calbiochem, CA, USA) for 1 h at 4 °C. After centrifugation (12,000×$g$ × 10 min), supernatants were mixed with an equal volume of sample buffer (62.5 mM Tris-HCl (pH 6.8), 2% (w/v) SDS, 2.5% (v/v) 2-mercaptoethanol, 5% (v/v) glycerol, and 0.0025% (w/v) bromophenol blue). The BCA method (Pierce BCA Protein Assay Kit; Thermo Scientific, IL, USA) was used to determine protein concentrations, which were subsequently equalized across samples. Samples were separated by SDS-PAGE, transferred onto polyvinylidene difluoride membrane Immobilon-P (Millipore) by the semi-dry method, blocked with 5% milk in TBST (10 mM Tris/HCl (pH 8.0), 150 mM NaCl, 0.05% Tween-20), and reacted with the following primary and secondary antibodies diluted in TBST with 0.1% skim milk or Can Get Signal solution (Toyobo, Osaka, Japan): rabbit anti-YAPdeltaC[13], 1:20,000 (raised against the common COOH-terminal peptide [SVFSRDDSGIEDNDNQ]); rabbit anti-FLAG, 1:2000 (F7425, SIGMA, IL, USA); mouse anti-c-Myc, 1:2000, (#sc-40 Santa Cruz Biotechnology, Dallas, TX, USA); rabbit anti-RORα, 1:1000 (#sc-28612, Santa Cruz Biotechnology); mouse anti-RGS-His, 1:5000 (#34650, Qiagen, Hilden, Germany); rabbit anti-GST, 1:3000 (sc-469, Santa Cruz Biotechnology); rabbit anti-YAP, 1:5000 (#14074S, Cell Signaling Technology, MA, USA); rabbit anti-Tip60, 1:5000 (#PA5-23290, Thermo Scientific); anti-Atxn1, 1:2000 (#MABN37, Millipore, MA, USA); mouse anti-RpA1, 1:1000 (H-7, Santa Cruz Biotechnology), anti-γH2AX, 1:1000 (Ser139, #05-636, Millipore); anti-53BP1, 1:10,000 (NB100-304, Novus Biologicals, CO, USA): anti-GAPDH, 1:5000 (MAB374, Millipore, MA, USA); anti-α-tubulin, 1:3000 (T6199, Millipore); anti-β-actin, 1:1000 (sc-47778, Santa Cruz Biotechnology); HRP-linked anti-rabbit IgG, 1:3000 (NA934, GE Healthcare, IL, USA); HRP-linked anti-mouse IgG, 1:3000 (NA931, GE Healthcare). Primary and secondary antibodies were incubated overnight at 4 °C and for 1 h at room temperature (RT), respectively. ECL Prime Western Blotting Detection Reagent (RPN2232, GE Healthcare) and a luminescent image analyzer (ImageQuant LAS 500, GE Healthcare) were used to detect proteins. Uncropped western blots are shown in Supplementary Fig. 9.

**Immunoprecipitation**. 293T or COS-7 cells were transfected with the indicated plasmids and harvested 48 h after transfection. Cells were lysed with TNE buffer (10 mM Tris-HCl (pH 7.5), 150 mM NaCl, 1 mM EDTA, 1% Nonidet P-40).

Lysates were collected by centrifugation (15,000×g × 10 min). Aliquots were incubated for 2 h with a 50% slurry of Protein G–Sepharose beads (GE Healthcare). After centrifugation (2000×g × 3 min), the supernatants were incubated with 1 µg of rabbit anti-FLAG antibody (#F7425, SIGMA, St. Louis, MO, USA), mouse anti-Myc antibody (#sc-40, Santa Cruz Biotechnology), or rabbit anti-RORα antibody (#sc-28612, Santa Cruz Biotechnology) overnight at 4 °C; incubated with Protein G–Sepharose beads for 2 h; washed with TNE buffer; and eluted with sample buffer.

**Immunohistochemistry**. Mouse brains were fixed in 4% paraformaldehyde for 12 h. Paraffin sections (thickness, 5 µm) were de-paraffinized in xylene, re-hydrated, dipped in 0.01 M citrate buffer (pH 6.0), and microwaved at 120 °C for 15 min. After blocking with 10% FBS containing PBS, sections were incubated with primary antibody for 12 h at 4 °C, washed with PBS three times at RT, and incubated with secondary antibodies at RT for 1 h. All procedures were performed in parallel for all mouse groups being compared.

The antibodies used for immunohistochemistry were diluted as follows: rabbit anti-calbindin D-28K antibody, 1:2000 (C2724, Sigma); mouse anti-calbindin D-28K antibody, 1:2000 (C9848, Sigma); rabbit anti-YAP antibody, 1:100 (#sc-15407, Santa Cruz Biotechnology); rabbit anti-YAPdeltaC antibody, 1:1000 (raised against the common COOH-terminal peptide [SVFSRDDSGIEDNDNQ]); rabbit anti-RORα, 1:100 (#sc-28612, Santa Cruz Biotechnology); rabbit anti-Tip60, 1:100 (#sc-25378, Santa Cruz Biotechnology); mouse anti-Atxn1, 1:100 (#MABN37, Millipore); anti-γH2AX, 1:100 (#05-636, Millipore); rabbit anti-53BP1, 1:5000 (NB100-304, Novus Biologicals); mouse anti-RpA1, 1:100 (H-7, Santa Cruz Biotechnology, CA, USA); Cy3-conjugated anti-rabbit IgG, 1:500 (711-165-152, Jackson Laboratory, Bar Harbor, ME, USA); Alexa Fluor 488–conjugated anti-mouse IgG, 1:1000 (A21202, Molecular Probes, Eugene, OR, USA); Cy3-conjugated anti-mouse IgG, 1:500 (715-165-150, Jackson Laboratory); Alexa Fluor 488-conjugated anti-rabbit IgG, 1:1000 (A21206, Molecular Probes).

**Confocal microscopy analysis**. Fluorescence confocal images were acquired on a laser scanning confocal microscope (LSM510META, Carl Zeiss, Germany; UPLANSAPO ×10, ×40, and ×60; multiline Argon and HeNe(G) lasers; three channels). Acquired images were analyzed using the ImageJ software (http://imagej.nih.gov/ij/) for measurements of signal intensity. The mean values of nuclear signals were compared among three groups (C57BL/6, *Atxn1*-KI, and Group I).

**Luciferase assay**. The following plasmids (0.5 µg) were transiently transfected into COS-7 cells ($1 \times 10^5$ cells/well) using Lipofectamine 2000 (Invitrogen, MA, USA), and luciferase assays were performed using the Dual-Glo Luciferase assay system (Promega, WI, USA) 24 h after transfection. Effector plasmids were RORα-pCIneo, RORα-Y507A-pCIneo, full-length YAP2-pCIneo, and YAPdeltaC-pCIneo, and the reporter plasmid was *NRD1*-RORE-pGreenfire.

**Pull-down assay**. GST-RORα (human partial RORα, a.a. 424–523; Abnova, Taipei, Taiwan), His-Tip60 (Cat#10783, Cayman Chemical, Ann Arbor, MI, USA) and His-YAP (MBS717875, MyBioSource, CA, USA) proteins were mixed in 400 µl of reaction solution (PBS with 0.1% Tween-20 and the protease inhibitor cocktail) and incubated for 12 h at 4 °C. After addition of 50 µl of 50% glutathione–Sepharose suspended in reaction solution (Cat#17513201, GE Healthcare), the mixture was incubated for another 3 h at 4 °C, centrifuged at 500×g for 1 min, washed with PBS three times, and solubilized in sample buffer. The samples were then subjected to western blotting. Primary and secondary antibodies were as follows: rabbit anti-YAP (#14074, Cell Signaling Technology, MA, USA); mouse anti-RGS-His (34650, Qiagen); rabbit anti-GST (sc-459, Santa Cruz Biotechnology); HRP-linked anti-mouse IgG, 1:3000 (NA931, GE Healthcare); and HRP-linked anti-rabbit IgG, 1:3000 (NA934, GE Healthcare). Primary and secondary antibodies were incubated overnight at 4 °C and for 1 h at room temperature, respectively. ECL Prime Western Blotting Detection Reagent and an ImageQuant LAS 500 luminescent image analyzer were used to detect proteins.

**ChIP assay**. ChIP assays were performed using the SimpleChIP plus Enzymatic Chromatin IP kit (Cell Signaling Technology, #9005) following the manufacturer's protocol with some modifications. In brief, mouse cerebellar tissues (10 mg) were minced in 1 ml of ice-cold PBS; 42.5 µl of 37% formaldehyde was added, and the mixture was incubated for 20 min at room temperature to allow cross-linking. After addition of glycine to stop the cross-linking reaction, the suspended tissues were centrifuged, washed twice with ice-cold PBS, homogenized using a type B Dounce homogenizer, re-suspended in kit Buffer A to perforate the cell membrane, and incubated for 20 min at 37 °C with micrococcal nuclease. Nuclei were destroyed by sonication, and the debris was removed by centrifugation. The clarified nuclear extracts were incubated overnight at 4 °C with anti-YAP antibody or anti-YAPΔC antibody and immunoprecipitated with protein G magnetic beads; the resultant protein–DNA complex was subjected to PCR amplification of the RORE in promoters of target genes. PCR was performed using PrimeSTAR HS DNA polymerase (R010A, Takara Bio Inc., Shiga, Japan). Forward and reverse primers for amplification of RORE were used as follows: *Pcp2*, 5′-

CAGTCCTTAACCTGCAAGGC-3′ and 5′-CCTGGAACTCCTGCTGTCAT-3′; *Pcp4*, 5′-AATCAACAACCCTCGCTGTC-3′ and 5′-GTTTGGGGTCACCA-TAGCTT-3′; *Slc1A6* 5′-GGGCATGTGATTCAGTTGGT-3′ and 5′-GCTTGTGAGCTCTTGTGCAG-3′. PCR conditions were as follows: 35 cycles of 98 °C for 10 s (denaturation), 65 °C for 15 s (annealing), and 72 °C for 60 s (extension).

**Quantitative RT-PCR**. Total RNA from mouse cerebellum was purified using the RNeasy mini kit (Qiagen, Limburg, Netherlands). Purified total RNA was reverse-transcribed using SuperScript VILO (Invitrogen). Quantitative PCR analyses were performed on a 7500 Real-Time PCR system (Applied Biosystems, Foster City, CA, USA) using the Thunderbird SYBR Green (Toyobo, Osaka, Japan). The primer sequences for RORα target genes, based on information in a previous report, were as follows[24]:

Mouse *A2bp1*: forward, 5′-AGACCACTGTCCCTGACCAC-3′; reverse, 5′-CATTTGTCGGAGGTCTGGAT-3′.

Mouse *Cyp19a1*: forward, 5′-CTTTCAGCCTTTTGGCTTTG-3′; reverse, 5′-ATTTCCACAAGGTGCCTGTC-3′.

Mouse *Ctgf*: forward, 5′-TGCGAAGCTGACCTGGAGGAAA-3′; reverse, 5′- CCGCAGAACTTAGCCCTGTATG-3′.

Mouse *Cyr61*: forward, 5′-GTGAAGTGCGTCCTTGTGGACA-3′; reverse, 5′- CTTGACACTGGAGCATCCTGCA-3′.

Mouse *Ankrd1*; forward, 5′-GCTTAGAAGGACACTTGGCGATC-3′; reverse, 5′-GACATCTGCGTTTCCTCCACGA-3′.

Mouse *Bcl-2*; forward, 5′-CCTGTGGATGACTGAGTACCTG-3′; reverse, 5′- AGCCAGGAGAAATCAAACAGAGG-3′.

Mouse *c-Flip*; forward, 5′-GCTCTACAGAGTGAGGCGGTTT-3′; reverse, 5′- CACCAATCTCCATCAGCAGGAC-3′.

Mouse *XIAP*; forward, 5′- GGCAGAATATGAAGCACGGATCG-3′; reverse, 5′- CACTTGGCTTCCAATCCGTGAG-3′.

Mouse *Naip1*; forward, 5′- CGAGGTCTCAGAGACAAACCAG-3′; reverse, 5′- GAACTCTCCAGGAAGGACTGAG-3′.

**Statistics**. One-way ANOVA followed by Tukey's HSD test was used for comparisons among multiple groups. We performed power analysis to estimate the required sample size ($n$) for each experiments. All data are shown as mean ± SEM.

**Data availability**. The authors declare that the data supporting the findings of this study are available within the article and supplementary information or available from the corresponding author upon request.

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

## Acknowledgements

This work was supported by a Grant-in-Aid for Scientific Research on Innovative Areas (Foundation of Synapse and Neurocircuit Pathology, 22110001 and 22110002) from the Ministry of Education, Culture, Sports, Science and Technology of Japan, a Grant-in-Aid for Scientific Research (A) (16H02655) from Japan Society for Promotion of Science (JSPS), partly Brain Mapping by Integrated Neurotechnologies for Disease Studies (Brain/MINDS) and Strategic Research Program for Brain Sciences (SRPBS) from Japan Agency for Medical Research and Development (AMED), a CREST grant from Japan Science and Technology Agency (JST) and grants for research on intractable diseases from AMED (to H.O.) We thank Dr Huda Y. Zoghbi (Baylor College of Medicine) for *Atxn1*-KI mice. We also thank Ms. Ayako Seki (TMDU) for manuscript preparation and Dr Hikaru Ito, Mr. Keisuke Kurosu, and Ms. Tayoko Tajima (TMDU) for technical assistance.

## Author contributions

K.F., Y.M. and S.U. performed experiments and wrote the paper. X.C., H.S., T.T., H.I., K. W., H.H. and K.T. performed experiments. M.S. prepared materials and designed research. H.O. planned the project, designed research, and wrote the paper.

## Additional information

**Competing interests:** The authors declare no competing financial interests.

