## [Peer Review File · Nature Communications]

Reviewers' comments:

Reviewer #1 (Remarks to the Author):

Mutant Atxn1, a SCA1 neurodegenerative disorder protein, decreases ROR α widely affects expression of target genes necessary for the cerebellar development. The authors previously showed that cerebral cortex neuron expressing YAP isoform YAPdeltaC is resistant to transcriptional repression-induced a typical cell death (TRIAD) induced by alpha-amanitin that might be relevant to neurodegeneration. In current study, they showed that developmental but not adulthood expression of YAPdeltaC rescues neurodegeneration phenotypes of mutant ataxin-1 knock-in (Atxn1-KI) mice. YAPdeltaC was found to interact with ROR α and enhance ROR α transcriptional activity. Mutant Atxn1 deprives YAPdeltaC from the transcriptional complex of ROR α to suppress transcription. Mutant Atxn1 also decreases ROR α and YAPdeltaC via protein degradation. Genetic supplementation of YAPdeltaC recovers protein levels of ROR α and YAPdeltaC and restores transcription of target genes of ROR α in Atxn1-KI mice.

Major concerns:

1. Since YAP is an effective anti-death effector, YAP overexpression likely makes neuronal cell more resistant to mutant ataxin-1-induced cell death by upregulating anti-apoptotic genes independent of ROR α . The author should check their expression and address this possibility.
2. The authors should explain why mutant but not WT ataxin-1 deprives YAPdeltaC from ROR α .
3. Overexpression of YAP commonly induces tumor formation and cell de-differentiation so that it is important for the authors to examine the effect of different expression levels of YAPdeltaC on neuronal cells.
4. Mutant Atxn1 also decreases ROR α and YAPdeltaC via protein degradation in a long time span in vitro. This could be due to more number of unhealthy or dying cells induced by expression of mutant Atxn1. The author should address this possibility.
5. Most data are from 293T or Cos-7 cell lines. The authors should repeat the experiments in primary neuron cells.

Reviewer #2 (Remarks to the Author):

Here the investigators further examine the molecular basis for the previously developmental aspects of disease in SCA1 mice in relation to the transcriptional regulator ROR α . In earlier work this group explored the role of another transcriptional co-activator, YAP, in an atypical necrosis called TRIAD, transcriptional repression-induced atypical cell death. Building on the finding that TRIAD contributes to HD pathogenesis, they develop a dox-on conditional YAPdeltaC mouse and show that enhanced YAPdeltaC expression beginning at E0 suppressed motor deficits and improved survival of Sca1154Q/2Q knockin mice. In contrast, activation of YAPdeltaC at p21 failed to provide any sign of rescuing disease phenotypes. These results support one of their major conclusions – that YAPdeltaC expression dampens the developmental aspects of SCA1 pathogenesis. They further show that YAPdeltaC interacts with the transcriptional activator ROR α forming a complex with wt as well as expanded Atxn1. In an extensive series of tissue culture experiment using for the most part 293 cells these investigators examine molecular aspects of the YAPdeltaC-ROR α -Atxn1 complex and its ability to regulate transcription. This work shows that all components are needed for activity and that expanded Atxn1 disrupts complex formation/function. This disruption by expanded Atxn1 is overcome by enhanced expression of YAPdeltaC. In mice they examined the expression of the components in Purkinje cells of the complex by immunostaining and expression of two target ROR α -regulated genes by PCR. Overall this is an interesting study that nicely expands understanding of developmental aspects of SCA1 pathogenesis. However, there are some issues that if addressed would strengthen this study.

1. The extension of survival of Sca1154Q/2Q mice in YAPdeltaC activated mice is very intriguing. To my knowledge this is the most dramatic demonstration of an effect on survival to date. Yet there is no discussion on this finding in relation to TRIAD/RORa in regions outside of the cerebellum that might underlie premature lethality in the Sca1154Q/2Q knockin mice.
2. A minor issue is that the authors should consider moving some of the culture data to supplementary material in order that the YAPdeltaC-Atxn1-RORa co-expression might be moved from the supplementary data to body of the manuscript.
3. The authors need to correct instances where the previous developmental work in SCA1 pathogenesis is cited involving the Sca1154Q/2Q knockin mice. The earlier study used ATXN1 transgenic where transgene expression was limited to Purkinje cells. This brings up the point that they might consider discussing the fact that a very similar role for cerebellar development effect is shared between ATXN1 Purkinje cell transgenic mice and the Sca1154Q/2Q knockin mice.

Reviewer #3 (Remarks to the Author):

The study by Fujita and colleagues analyzes YapdeltaC in the context of SCA1. The group previously identified a protective role of the C-terminal truncated isoforms of Yap. The protective role is under the setting of a form of cell death that only this group has reported on. In this work, they test the effects of YapdeltaC on SCA1 progression by generating a dox-inducible YapdeltaC line, crossing that to the KI, and testing how induction at E0 vs 8-wk old vs no induction.

Overall the data are interesting. There is a clear impact of expressing more YapdeltaC at birth. There is an improvement in motor behavior and pathological readouts. The animals are not normal, however. This work is very similar to an earlier report by this group, where YapdeltaC's impact in HD model systems was evaluated

The in vitro data show that Atx1 and RORa bind YAP, forming a ternary complex, and that mutant atxn1 also binds YapdeltaC. To conclude that these studies show that there is a functional balance between Yap/YapdeltaC and mutant Atx1 in RORalpha-mediated transduction during development that destines the adulthood pathology of SCA1 (last sentence in abstract and stated elsewhere, including title) is quite a stretch.

Developmental effect of YAPdeltaC on Atxn1 mice pathology: Its not mentioned from the write up which lobules were examined, how many regions/cortex/animal, and if there was any variation among the various lobules as to the extent of the recovery. Also not clear how many cells/lobule were counted and how many sections were analyzed per animal per lobule.

The authors use Cos7 cells, HEK293 cells, and perform in vivo studies. The rationale for the various in vitro studies, which appear confirmatory to what was already published by this group, should be placed into the supplemental material. Also, for the in vitro studies, gross overexpression often occurs. It would be important to confirm in vitro findings in vivo. Obviously this cannot be done with the stoichiometry studies. But for the ChIP and the studies in Fig 7, the in vitro studies may be misleading as the authors even suggest.

The manuscript would benefit from serious editing for improving grammar and word usage.

RESPONSE TO REFEREES:

Reviewers' expertise:

Reviewer #1: Hippo signaling;

Reviewer #2: SCA1 mouse models, transcriptional control;

Reviewer #3: SCA1 mouse models, molecular and cell biology.

Reviewers' comments:

Reviewer #1 (Remarks to the Author):

Mutant Atxn1, a SCA1 neurodegenerative disorder protein, decreases RORa widely affects expression of target genes necessary for the cerebellar development. The authors previously showed that cerebral cortex neuron expressing YAP isoform YAPdeltaC is resistant to transcriptional repression-induced a typical cell death (TRIAD) induced by alpha-amanitin that might be relevant to neurodegeneration. In current study, they showed that developmental but not adulthood expression of YAPdeltaC rescues neurodegeneration phenotypes of mutant ataxin-1 knock-in (Atxn1-KI) mice. YAPdeltaC was found to interact with RORa and enhance RORa transcriptional activity. Mutant Atxn1 deprives YAPdeltaC from the transcriptional complex of RORa to suppress transcription. Mutant Atxn1 also decreases RORa and YAPdeltaC via protein degradation. Genetic supplementation of YAPdeltaC recovers protein levels of RORa and YAPdeltaC and restores transcription of target genes of RORa in Atxn1-KI mice.

>>> We thank very much the reviewer for the deep reading and correct understanding. We appreciate thoughtful and valuable comments and advices.

Major concerns:

1. Since YAP is an effective anti-death effector, YAP overexpression likely makes neuronal cell more resistant to mutant ataxin-1-induced cell death by

upregulating anti-apoptotic genes independent of RORa. The author should check their expression and address this possibility.

>>> We appreciate that the reviewer kindly referred and carefully considered about our previous work. We reported atypical cell death TRIAD (Hoshino et al, JCB2006) that belongs to type 3 necrosis and is independent of apoptosis. In this case, we could not find caspase activation, DNA fragmentation or cytochrome C release but found downregulation of YAP. Our following studies revealed that functional deficiency of YAP contributes to TRIAD (Mao et al, Hum Mol Genet 2016; Mao et al, Cell Death Dis 2016).

>>> Following the advice of the reviewer, we checked whether YAPdeltaC affected the expression of representative anti-apoptotic genes in mice in vivo at P7, P21 and 32 weeks of Control, mutant Atxn1-KI and Group 1 mice (Figure 9). The analysis did not detect the increase of anti-apoptotic genes in their expression levels (Figure 9). One exception was Naip1, but it was already increased in Atxn1-KI mice without induction of YAPdeltaC (Figure 9). This result again is not supportive for apoptosis but may be suggestive for TRIAD. Consistently with this idea, we found suppression of TEAD/YAP-target genes in Atxn1-KI mice and their recovery in Group I mice (Figure 9).

>>> We newly discussed about relationship of the two types of cell death (apoptosis and TRIAD) to the delayed degeneration in SCA1 pathology in Discussion. In brief, we consider that our data do not support suppression of the apoptosis by YAP or YAPdeltaC contribute to the adult onset pathology. It seems definite that their effects on the RORa transcription target genes contribute to the adult onset pathology and cell death, given that RORa effect on the delayed pathology was shown by a previous report with Atxn1 transgenic mice (not KI) by Serra et al (Cell 2006). Meanwhile, with the results from new experiment suggested by the reviewer (Figure 9), we consider that delayed effect of YAP/YAPdeltaC on TRIAD (TEAD-YAP transcription mediated) after birth might function in parallel as the second downstream pathway of YAP/YAPdeltaC. The detailed analyses should be performed in a future study.

2. The authors should explain why mutant but not WT ataxin-1 deprives YAPdeltaC from RORa.

>>> We thank the reviewer for this critical question. This is actually a very important issue, we have noticed.

>>> In this version we showed that YAP/YAPdeltaC are degraded by proteasome after interaction with mutant Atxn1 (Figure 7C, Supplementary Figure 3C). The proteasome dependent degradation did not occur in YAP/YAPdeltaC-normal Atxn1 complex (Figure 7C, Supplementary Figure 3C). This mechanism will be one reason for why YAPdeltaC is deprived from RORa only by its binding to mutant Atxn1.

>>> The second reason will be impaired nuclear dynamics of YAP/YAPdeltaC by interaction with mutant Atxn1, which we have shown in other multiple molecules binding to mutant Atxn1 (Fujita et al, Nat Commun 2014; Taniguchi et al, Hum Mol Genet 2016). Due to the slowed dynamics (or aggregation-prone nature) of mutant Atxn1 protein, the interacting partner is generally disturbed in cellular dynamics (Fujita et al, Nat Commun 2014; Taniguchi et al, Hum Mol Genet 2016).

In this specific case, the place in the nucleus for interaction between YAP/YAPdeltaC and Atxn1 and the place in the nucleus for interaction between RORa and YAP/YAPdeltaC will be different, as known in factors for basic nuclear machineries (ref. Gall et al, 1999), and the impaired dynamics might disturb formation of the complex of three proteins. This explanation remains in hypothesis. However, since basic concept has been proven in our previous studies (the repetition of our previous result is suggested to be excluded from this paper by a reviewer) and since we need to do a large number of experiments in order to strictly prove it, we believe that these substantial amounts of experiments should be analyzed in a next paper.

In this paper, instead, we have made a new section in Discussion and argued about the possible second mechanism.

3. Overexpression of YAP commonly induces tumor formation and cell de-differentiation so that it is important for the authors to examine the effect of different expression levels of YAPdeltaC on neuronal cells.

>>> We checked the morphology of primary cerebellar neurons in which YAP or YAPdeltaC were expressed at multiple levels (Supplementary Figure 7A). There was no morphological change in such neurons (Supplementary Figure 7B). There was no increase of Ki67 or Sox2 at morphological or WB levels (Supplementary Figure 7C, D), excluding proliferation and de-differentiation.

4. Mutant Atxn1 also decreases RORa and YAPdeltaC via protein degradation in a long time span in vitro. This could be due to more number of unhealthy or dying cells induced by expression of mutant Atxn1. The author should address this possibility.

>>> We showed that expression levels of multiple control genes such as GAPDH, tubulin and actin remain constant after transfection of mutant Atxn1 until day 7, indicating that the mutant Atxn1-transfected 293T cells were healthy until analyzed for RORa and YAPdeltaC levels (Supplementary Figure 3C). In addition, we showed phase contrast images after transfection of mutant Atxn1 from day 1 to day 7, which support healthy condition of mutant Atxn1-expressing cells (Supplementary Figure 3D).

>>> Similarly, we similarly checked the viability of primary cerebellar neurons at 7 days after transfection by the control gene expression levels (Figure 7C), the morphological level (Figure 7D) and quantitative data of % trypan blue-positive cells (Figure 7E). These results suggested no increase of cell death or of vulnerable cells by mutant Atxn1. These results are consistent with the fact that major groups of SCA1 research and also our group have not observed definite increase of cell death by expression of mutant Atxn1 in primary neurons and culture cell line.

>>> The negative cell death induction looks contrastive to the cell death by expression of mutant Huntingtin Exon1 in primary neurons and culture cell line. However, the process to cell death and necessary time for cell death to occur might be different between HD and SCA1 pathologies, as we and other groups reported (for instance, Shiraishi et al, 2014, <https://doi.org/10.1371/journal.pone.0116567>).

5. Most data are from 293T or Cos-7 cell lines. The authors should repeat the experiments in primary neuron cells.

>>> Following the advice of the reviewer, we used primary cerebellar neurons, and repeated similar experiments (Figure 3D, E, Figure 4A, B, Figure 6C, D, Figure 7A, B, C, D, E Supplementary Figure 7A, B, C, D). Based on the data from primary neurons, we reconfirmed our previous conclusion that had been deduced from the data with cell lines.

Reviewer #2 (Remarks to the Author):

Here the investigators further examine the molecular basis for the previously developmental aspects of disease in SCA1 mice in relation to the transcriptional regulator ROR alpha. In earlier work this group explored the role of another transcriptional co-activator, YAP, in an atypical necrosis called TRIAD, transcriptional repression-induced atypical cell death. Building on the finding that TRIAD contributes to HD pathogenesis, they develop a dox-on conditional YAPdeltaC mouse and show that enhanced YAPdeltaC expression beginning at E0 suppressed motor deficits and improved survival of Sca1154Q/2Q knockin mice. In contrast, activation of YAPdeltaC at p21 failed to provide any sign of rescuing disease phenotypes. These results support one of their major conclusions – that YAPdeltaC expression dampens the developmental aspects of SCA1 pathogenesis. They further show that YAPdeltaC interacts with the transcriptional activator RORalpha forming a complex with wt as well as expanded Atxn1. In an extensive series of tissue culture experiment using for

the most part 293 cells these investigators examine molecular aspects of the YAPdeltaC-RORa-Atxn1 complex and its ability to regulate transcription. This work shows that all components are needed for activity and that expanded Atxn1 disrupts complex formation/function. This disruption by expanded Atxn1 is overcome by enhanced expression of YAPdeltaC. In mice they examined the expression of the components in Purkinje cells of the complex by immunostaining and expression of two target RORa-regulated genes by PCR. Overall this is an interesting study that nicely expands understanding of developmental aspects of SCA1 pathogenesis. However, there are some issues that if addressed would strengthen this study.

>>> We thank very much for the kind and high evaluation of the reviewer. We also appreciate very much thoughtful and valuable comments and advices.

1. The extension of survival of Sca1154Q/2Q mice in YAPdeltaC activated mice is very intriguing. To my knowledge this is the most dramatic demonstration of an effect on survival to date. Yet there is no discussion on this finding in relation to TRIAD/RORa in regions outside of the cerebellum that might underlie premature lethality in the Sca1154Q/2Q knockin mice.

>>> This is a really important comment. We added discussions about the extra-cerebellar or extra-brain role of YAPdeltaC in TRIAD/RORa. To discuss about this issue, the key point will be tissue or brain part distribution of Tet-ON and NSE (neuron-specific enolase) promoter-driven expression of YAPdeltaC in Tg mice. NSE gene expression was summarized from database:

Genecard

<http://www.genecards.org/cgi-bin/carddisp.pl?gene=ENO2>

MGI

<http://www.informatics.jax.org/gxd>

<http://www.informatics.jax.org/gxd#gxd=nomenclature%3DNSE%26vocabTerm%3D%26annotationId%3D%26locations%3D%26locationUnit%3Dbp%26struct>

ure%3D%26structureID%3D%26theilerStage%3D0%26results%3D100%26star
tIndex%3D0%26sort%3D%26dir%3Dasc%26tab%3Dresultstab

From these databases, the expression of NSE is extremely lower in the outside of brain.

>>> Also from our previous results using the same NSE-driven (the same promoter driven) HMGB1 transgenic mice (Ito et al, Embo Mol Med 2016), we can speculate gene expression patterns driven by the NSE promoter. Though the expression was detected in various parts of the brain, the expression levels in Purkinje cells is substantially higher than other neurons in the other regions (Ito et al, Embo Mol Med 2016). We had also checked the expression in other organs but the level was almost zero (data not shown). Therefore, though we cannot completely exclude the possibility that the effects of TRIAD/RORa in the other organs, other brain regions or in the other cells, we can speculate that a large part of the effect of YAPdeltaC on the elongated survival of Group I mice would be derived from cerebellar cells or Purkinje cells. We discussed about this issue in Discussion.

2. A minor issue is that the authors should consider moving some of the culture data to supplementary material in order that the YAPdelataC-Atxn1-RORa co-expression might be moved from the supplementary data to body of the manuscript.

>>> We moved culture cell-based data in old Figure 3, 4, 6 and 7 to Supplementary materials. Instead, we added the similar data newly obtained from primary neurons (Figure 3D, E, Figure 4A, B, Figure 6C, D, Figure 7A, B, C, D, E Supplementary Figure 7A, B, C, D), to respond to the comments from reviewer #1 and #3.

Following the advice of reviewer #2, we also moved some core data on the YAPdelataC-Atxn1-RORa co-expression from Supplementary Figure 2, 3, 4, to new Figure 8. We also added corresponding data of mice at P7 and made new Figure 8.

3. The authors need to correct instances where the previous developmental work in SCA1 pathogenesis is cited s involving the Sca1154Q/2Q knockin mice. The earlier study used ATXN1 transgenic where transgene expression was limited to Purkinje cells. This brings up the point that they might consider discussing the fact that a very similar role for cerebellar development effect is shared between ATXN1 Purkinje cell transgenic mice and the Sca1154Q/2Q knockin mice.

>>> We thank the reviewer very much for correcting our misunderstanding. We followed the advice and discuss about the similar effects on Purkinje cells of Tg mice and of KI mice (page 13 and page 18).

Reviewer #3 (Remarks to the Author):

The study by Fujita and colleagues analyzes YapdeltaC in the context of SCA1. The group previously identified a protective role of the C-terminal truncated isoforms of Yap. The protective role is under the setting of a form of cell death that only this group has reported on. In this work, they test the effects of YapdeltaC on SCA1 progression by generating a dox-inducible YapdeltaC line, crossing that to the KI, and testing how induction at E0 vs 8-wk old vs no induction.

Overall the data are interesting. There is a clear impact of expressing more YapdeltaC at birth. There is an improvement in motor behavior and pathological readouts. The animals are not normal, however. This work is very similar to an earlier report by this group, where YapdeltaC's impact in HD model systems was evaluated

The in vitro data show that Atx1 and RORa bind YAP, forming a ternary complex, and that mutant atxn1 also binds YapdeltaC. To conclude that these studies

show that there is a functional balance between Yap/YapdeltaC and mutant Atx1 in RORalpha-mediated transduction during development that destines the adulthood pathology of SCA1 (last sentence in abstract and stated elsewhere, including title) is quite a stretch.

>>> We appreciate the criticism of the reviewer and we deleted such sentences from Abstract or Results/Discussion.

Developmental effect of YAPdeltaC on Atxn1 mice pathology: Its not mentioned from the write up which lobules were examined, how many regions/cortex/animal, and if there was any variation among the various lobules as to the extent of the recovery. Also not clear how many cells/lobule were counted and how many sections were analyzed per animal per lobule.

>>> We observed primary fissure sides of lobule IV-V in Vermis, which corresponds to upper Vermis affected severely in SCA1 human patients (Rüb et al, Neuropath Applied Neurobiol 2012). The number of Purkinje cells in 600µm of primary fissure side of lobule IV-V was counted in 1 slide, and the values of 20 slides from 4 mice (5 slides from 1 mouse) were used for statistical analysis. For thickness of molecular layer, the thickness was measured at 6 regions in primary fissure side of lobule IV-V per slide, and the values of 120 regions from 20 slides of 4 mice were used statistical analyses. The diameter of Purkinje was measured in all Purkinje cells on primary fissure side of lobule IV-V, and the values from 20 slides of 4 mice were directly used for the analysis.

We wrote the details in Figure 2 legend. Regarding the difference among lobules, we performed similar analyses in lobule VII but the difference was not found in conclusion. We described about this in the text.

The authors use Cos7 cells, HEK293 cells, and perform in vivo studies. The rationale for the various in vitro studies, which appear confirmatory to what was already published by this group, should be placed into the supplemental material.

>>> We appreciate the comment regarding moving some figures to supplementary materials, which is similar to the suggestion from reviewer #2. We moved the old Figure 3, 4, 6, 7 to Supplementary materials.

Also, for the in vitro studies, gross overexpression often occurs. It would be important to confirm in vitro findings in vivo. Obviously this cannot be done with the stoichiometry studies.

>>> We accept completely the criticism. Overexpression is a critical experimental problem that has been recognized in the field. We added the experiments with cerebellar tissues (Figure 6A, B, Figure 8A, B, Figure 9) and re-examined our previous conclusions.

>>> Regarding in vivo interaction of YAP/YAPdeltaC-RORa and of YAP/YAPdeltaC-Atxn1, we added data at P7 of IP and ChIP with cerebellar tissues to Figure 6A, B, respectively. The in vivo IP data confirmed decreased interaction between YAP/YAPdeltaC and RORa in KI mice and recovery in Group I mice (Figure 6A). The in vivo ChIP data also confirmed decreased amounts of YAP/YAPdeltaC interacting with RORa on the cis-element of RORa-responsive genes in KI mice and the recovery in Group I mice (Figure 6B).

>>> In new Figure 8, we compared “in vivo” expression levels of related proteins by WB and immunohistochemistry with mouse cerebellar tissues at P7, P21 and 32 weeks. The data showed that RORa was not changed at P7 while it was decreased at P21 and 32 weeks. Decrease of YAP preceded RORa at P7. Here, we moved some data from old Supplementary Figure 2, 3, 4, added new data of WB and IHC at P7 and made new Figure 8.

>>> For more detailed interaction manners, we used primary cerebellar neurons, and performed IP (Figure 3D, E and Figure 4A, B).

>>> We performed qPCR to quantify target genes (and anti-apoptotic genes) in mouse cerebellar tissues (Figure 9).

But for the ChIP and the studies in Fig 7, the in vitro studies may be misleading as the authors even suggest.

>>> We wrote in the previous version “our finding in transient expression is obviously different from the reported decrease of RORa”. To reconcile the gap between in vivo study previously reported by other group and our in vitro result with HEK293T cells, we repeated the IP and ChIP experiments (Figure 6A, B) and the WB and IHC experiments (Figure 8A, B) by using cerebellar tissues from three genotypes of mice. We also performed IP and ChIP by using mouse cerebellar neurons (Figure 3D, E, Figure 4A, B, Figure 6C, D, Figure 7C). We think our new results from in vivo studies could reach to a hypothesis explaining both of the previous scheme reported from the other group and our scheme. RORa is decreased following YAP/YAPdeltaC with a time lag.

The manuscript would benefit from serious editing for improving grammar and word usage.

>>> We asked a professional editor to correct and improve English.

REVIEWERS' COMMENTS:

Reviewer #1 (Remarks to the Author):

I think that the authors did an impressive job answering all questions posed by the reviewer. I have one remaining comment and suggestion.

The mouse genetic analyses performed in this study revealed that expression of YAPdeltaC during development, but not in adulthood, markedly ameliorated the behavioral and pathological phenotypes of Atxn1-KI mice in adulthood. The authors demonstrated that YAP/YAPdeltaC-mediated ROR α transcription is critical for the phenotypic recovery. Likely, the YAP/YAPdeltaC/ROR α complex is required for neuronal differentiation and maturation. Recent study showed that TAZ, the mammalian homolog of YAP, functioned as a co-activator of ROR γ t to promote TH17 differentiation through similar biochemical mechanism (Nat Immunol. 2017 Jul;18(7):800-812.) .The authors should add the discussion about that.

Reviewer #2 (Remarks to the Author):

In this revised manuscript a strength remains the remarkable affect of early expression of YAP has on cerebellar/Purkinje cell pathology in Sca1-154Q knockin mice versus the lack of an affect with adult YAP expression. These clearly support the increasing appreciation for the role of developmental changes in "adult" onset neurodegenerative disease. In response to this reviewer's comments about CNS regions important for lethality in this mouse model of SCA1, the authors have added additional discussion that supports their conclusion that alterations in Purkinje cells contribute to lethality in this model. One additional point along this line would be for them comment on this point re SCAs in humans where it is thought that lethality is largely due to non-cerebellar pathology. For example, SCA6 the purest cerebellar SCA is also a disease where patients often have a normal life span. Perhaps aspects of lethality in this model are unique to the mouse model? Regardless, this revised manuscript is much improved with its grammar and word usage much clearer.

Reviewer #3 (Remarks to the Author):

no additional concerns

REVIEWERS' COMMENTS:

Reviewer #1 (Remarks to the Author):

I think that the authors did an impressive job answering all questions posed by the reviewer. I have one remaining comment and suggestion.

The mouse genetic analyses performed in this study revealed that expression of YAPdeltaC during development, but not in adulthood, markedly ameliorated the behavioral and pathological phenotypes of Atxn1-KI mice in adulthood. The authors demonstrated that YAP/YAPdeltaC-mediated ROR α transcription is critical for the phenotypic recovery. Likely, the YAP/YAPdeltaC/ROR α complex is required for neuronal differentiation and maturation. Recent study showed that TAZ, the mammalian homolog of YAP, functioned as a co-activator of ROR γ t to promote TH17 differentiation through similar biochemical mechanism (Nat Immunol. 2017 Jul;18(7):800-812.) .The authors should add the discussion about that.

>>> We thank the kind evaluation of reviewer #1. We followed the request, and added a paragraph in discussion to refer the suggested paper.

Reviewer #2 (Remarks to the Author):

In this revised manuscript a strength remains the remarkable affect of early expression of YAP has on cerebellar/Purkinje cell pathology in Sca1-154Q knockin mice versus the lack of an affect with adult YAP expression. These clearly support the increasing appreciation for the role of developmental changes in “adult” onset neurodegenerative disease. In response to this reviewer’s comments about CNS regions important for lethality in this mouse model of SCA1, the authors have added additional discussion that supports their conclusion that alterations in Purkinje cells contribute to lethality in this model. One additional point along this line would be for them comment on this point re

SCAs in humans where it is thought that lethality is largely due to non-cerebellar pathology. For example, SCA6 the purest cerebellar SCA is also a disease where patients often have a normal life span. Perhaps aspects of lethality in this model are unique to the mouse model? Regardless, this revised manuscript is much improved with its grammar and word usage much clearer.

>>> We thank the critical review of reviewer #2. The issue of non-neuronal or non-cerebellar pathologies is very important, and we recognize it. However, from the viewpoint of experienced neurology doctors, SCA1 patients mostly die due to pneumonia caused by dysphagia and aspiration, after they advance ataxic gait, become bed-ridden and suffer incoordination of pharyngeal and laryngeal muscles causing dysphagia and dysphasia (a part of ataxia). Though we had paragraphs about non-neuronal pathologies in previous version, we added another paragraph to discuss about it.

Reviewer #3 (Remarks to the Author):

no additional concerns

Thank you so much for kind effort of the reviewer #3.